# SurvivalPFN: Amortizing Survival Prediction via In-Context Bayesian Inference

**Shi-ang Qi** [1]   **Vahid Balazadeh** [1 2]   **Michael Cooper** [1 2]   **Russell Greiner** [3 4]   **Rahul G. Krishnan** [1 2]

## Abstract

Survival analysis models time-to-event outcomes under censoring, but selecting an appropriate estimator from many specialized approaches often requires substantial methodological expertise. We introduce SurvivalPFN, a prior-data fitted network that amortizes Bayesian inference for censored observations through in-context learning. SurvivalPFN is pretrained on a diverse family of synthetic, identifiable, and right-censored data-generating processes, enabling it to amortize survival analysis in a single forward pass during inference. As a result, the model adapts to the effective complexity of each dataset without task-specific training or hyperparameter tuning, avoids restrictive parametric assumptions, and produces calibrated survival distributions. In a benchmark spanning 61 survival datasets, 21 methods, and 5 evaluation metrics, SurvivalPFN achieves strong predictive performance and often improves upon established models. These results suggest that SurvivalPFN offers a principled and practical foundation model for survival analysis.

## 1. Introduction

Survival analysis models time-to-event outcomes under censoring. In right-censored data, the event time is only partially observed: for censored individuals, we know that the event has not occurred before the censoring time, but not when it would occur. Existing survival estimators address this partial observability with different assumptions, such as proportional hazards, parametric families, fixed time grids, or mixture-based likelihoods. These choices make practical survival modeling labor-intensive: practitioners must select, tune, validate, and often retrain a specialized estimator for each dataset.

[1]Vector Institute, Toronto, Canada [2]University of Toronto, Toronto, Canada [3]University of Alberta, Edmonton, Canada [4]Alberta Machine Intelligence Institute, Edmonton, Canada. Correspondence to: Shi-ang Qi <shiang.qi@vectorinstitute.ai>.

*Proceedings of the $2^{nd}$ ICML Workshop on Foundation Models for Structured Data*, Seoul, South Korea. 2026. Copyright 2026 by the author(s).

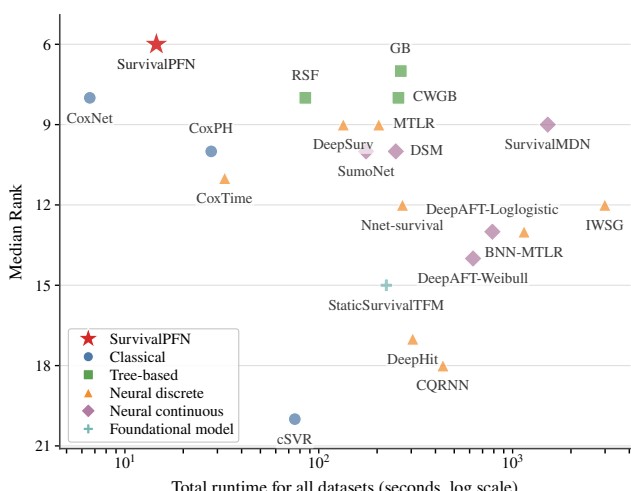

*Figure 1.* **Computational efficiency vs. performance across 61 datasets and 5 metrics.** SurvivalPFN achieves the best median rank while matching classical models in speed.

We introduce *SurvivalPFN*, a prior-data fitted network for amortized Bayesian survival prediction. SurvivalPFN is pretrained on millions of synthetic, identifiable, right-censored data-generating processes (DGPs). At test time, it receives an observed survival dataset as context and returns posterior predictive survival distributions for query individuals in a single forward pass. This shifts the cost of inference from dataset-specific optimization to an offline pretraining stage, while allowing the model to adapt in context to the amount and structure of evidence in each downstream dataset.

Figure 2 summarizes this contrast: classical survival analysis performs dataset-specific model selection and fitting to estimate $\widehat{S}(t \mid x)$, whereas SurvivalPFN amortizes this inference by learning a map from a censored context dataset $\mathcal{D}$ and query covariate $x^*$ to $\widehat{S}_\omega(t \mid x^*, \mathcal{D})$.

Our contributions are fourfold. First, we formulate survival prediction as amortized posterior predictive inference over identifiable right-censored DGPs. Second, we train a transformer to output event-time survival distributions directly from context data, avoiding task-specific tuning. Third, we provide a consistency argument showing that the Bayesian posterior predictive target recovers the true conditional survival function under identifiable censoring. Fourth, we evaluate SurvivalPFN on 61 held-out datasets against 21 baselines and find strong aggregate performance.

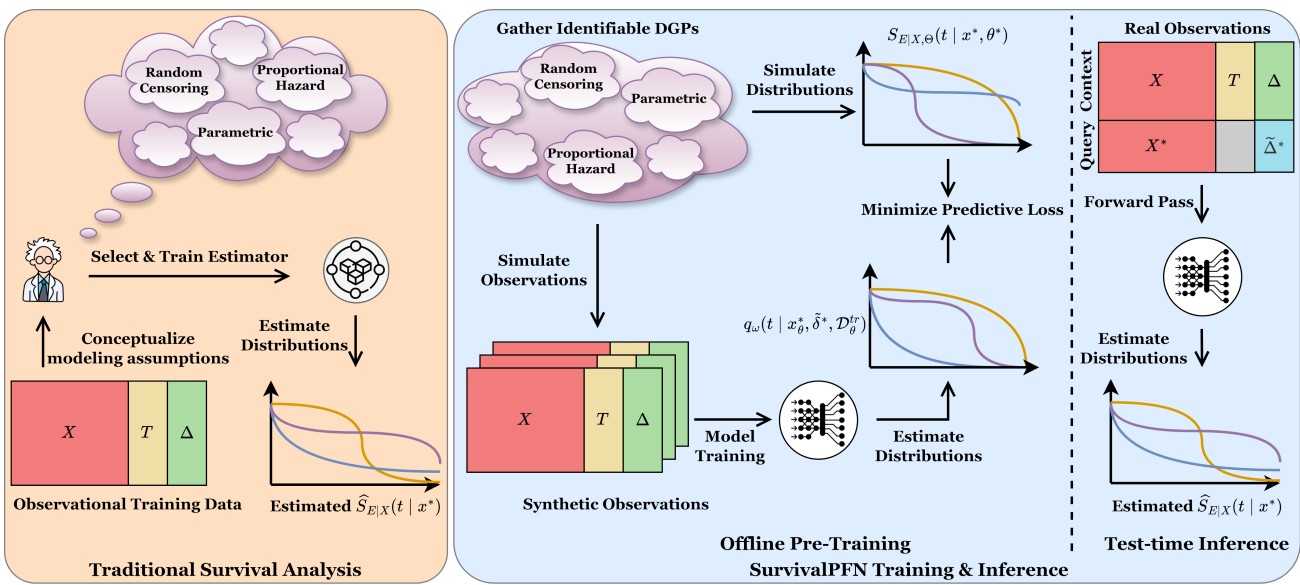

*Figure 2.* **Traditional survival analysis vs. SurvivalPFN.** *(Left):* Traditional survival analysis requires an analyst to select and fit a suitable estimator for the observational data. *(Right):* SurvivalPFN pre-trains on diverse synthetic, identifiable DGPs. At inference, an observed dataset is provided as context, and the survival distributions for query instances are obtained with a single forward pass.

## 2. Background

Let $X \in \mathbb{R}^d$ be covariates, $E \in \mathbb{R}_+$ the latent event time, $C \in \mathbb{R}_+$ the latent censoring time, $T = \min(E, C)$ the observed time, and $\Delta = \mathbb{1}[E \le C]$ the event indicator. Given observed data $\mathcal{D} = \{(x_i, t_i, \delta_i)\}_{i=1}^N$, the survival prediction target is the conditional event-time survival function

$$S_{E|X}(t \mid x) = \Pr(E > t \mid X = x).$$

Because $E$ and $C$ are not both observed, this target is not identifiable without assumptions. We assume conditional independent censoring, $E \perp C \mid X$, together with positivity over the evaluation time region. Under these assumptions, the conditional event-time distribution is identifiable from the observed law over $(X, T, \Delta)$; Appendix B gives the full argument and discusses non-identifiability under dependent censoring.

SurvivalPFN targets the Bayesian posterior predictive survival distribution induced by a prior over identifiable survival DGPs:

$$S_{E|X,\mathscr{D}}(t \mid x^*, \mathcal{D}) = \int_\Theta S_{E|X,\Theta}(t \mid x^*, \theta) f_{\Theta|\mathscr{D}}(\theta \mid \mathcal{D}) \, d\theta. \tag{2.1}$$

In ordinary Bayesian survival modeling, evaluating this quantity requires dataset-specific posterior inference. SurvivalPFN instead learns to approximate this map in context through prior-data pretraining.

## 3. Method

Figure 3 formalizes the SurvivalPFN pipeline summarized in Figure 2: synthetic DGPs generate context/query tasks, and $q_\omega$ is trained to return the requested event- or censoring-time distribution from $(\mathcal{D}, x^*, \widetilde{\delta}^*)$.

SurvivalPFN is a transformer $q_\omega$ that maps a right-censored context dataset and query covariates to a predictive distribution over time. Each context row $(x_i, t_i, \delta_i)$ is embedded as a context token. Each query token contains $x^*$ and a binary query indicator $\widetilde{\delta}^*$: setting $\widetilde{\delta}^* = 1$ asks for the event-time posterior predictive distribution, while $\widetilde{\delta}^* = 0$ asks for the censoring-time distribution. Context tokens attend to one another, while query tokens attend to the context but not to other queries, yielding permutation-invariant context processing and conditionally independent predictions across queries.

The model represents time with a high-resolution histogram over a monotone transformation of observed time. For event prediction, the predicted survival curve is the tail mass of the event-time histogram:

$$\widehat{S}_\omega(\tau_k \mid x^*, \mathcal{D}) = \sum_{\ell=k+1}^{L} q_{\omega,\ell}(x^*, \widetilde{\delta}^* = 1, \mathcal{D}). \tag{3.1}$$

Training follows the PFN principle: at each update, a synthetic identifiable DGP is sampled from the prior, a context/query task is generated, and the latent event or censoring time requested by $\widetilde{\delta}^*$ provides the supervised target. The loss is a discrete negative log likelihood over transformed time bins. Appendix D.1 details the synthetic

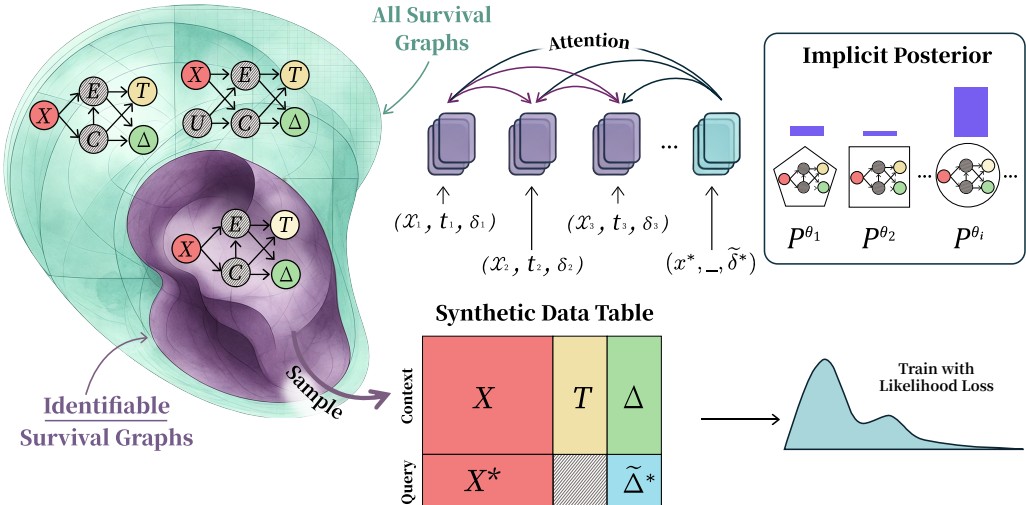

*Figure 3.* **Training SurvivalPFN.** At each iteration, we sample an identifiable survival DGP and use it to generate context tokens $(X, T, \Delta)$ together with query covariates $X^*$. Query tokens are formed by pairing $X^*$ with query indicators $\widetilde{\Delta}^*$, and SurvivalPFN predicts the requested event- or censoring-time distribution. The model is trained by minimizing the likelihood loss.

prior, Appendix D.2 describes architecture and inference, Appendix D.4 gives the full objective, and Appendix C states the posterior-predictive consistency result.

Our prior mixes flexible neural and distributional generators while enforcing conditional independent censoring by construction. The prior is designed to cover diverse covariate structures, event-time shapes, censoring rates, and observed-time dispersions.

## 4. Experiments and Results

We evaluate on 81 survival datasets, using 20 only for checkpoint selection and 61 as held-out benchmarks. Baselines span tabular foundation models, classical estimators, tree-based models, and neural discrete- and continuous-time models. We report IPCW-adjusted integrated Brier score (IBS), concordance index (CI), D-calibration, MAE, and log-rank reliability over 10 independent 70%/30% train/test splits. Figure 4 summarizes the dataset regimes, Appendix E.6 gives the full protocol, and Appendix F summarizes the datasets.

Figure 5 shows that SurvivalPFN achieves the strongest overall aggregate rank across the benchmark. It is especially strong on IBS, MAE, and log-rank reliability, indicating accurate probabilistic survival curves, useful time predictions, and agreement with observed time-to-event outcomes. It remains competitive on CI and D-calibration, where tree-based and Cox-style methods can also be strong.

The runtime comparison in Figure 1 shows that SurvivalPFN is only modestly slower than the fastest classical baseline while substantially improving aggregate rank. Additional

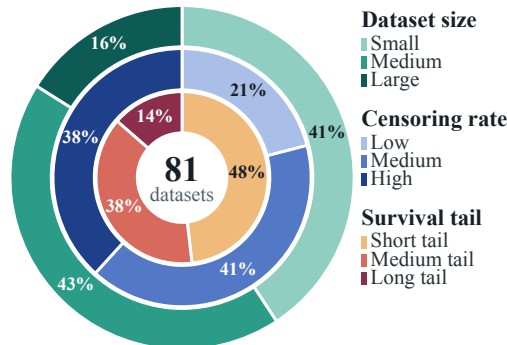

*Figure 4.* Summary of dataset size (cutoffs at 500 and 5000), censoring rate and tail rate (cutoffs at 33% and 67%).

results are deferred to the appendix: stratified performance by sample size and censoring rate appears in Appendix G.1; training-set-size sensitivity appears in Appendix G.3; comparison with general-purpose tabular foundation models appears in Appendix G.4; and ablations in Appendix G.5.

Several qualitative patterns are worth emphasizing. SurvivalPFN's strongest relative gains occur on smaller datasets, where conventional flexible models have limited data with which to estimate high-dimensional survival structure. This is the regime in which amortized Bayesian inference is most useful: the pretrained prior supplies an inductive bias, while the context set still determines the posterior predictive distribution for the current task. At the same time, SurvivalPFN remains competitive across censoring regimes, suggesting that the prior over identifiable right-censored DGPs captures a useful range of event/censoring interactions rather than overfitting to a single censoring pattern.

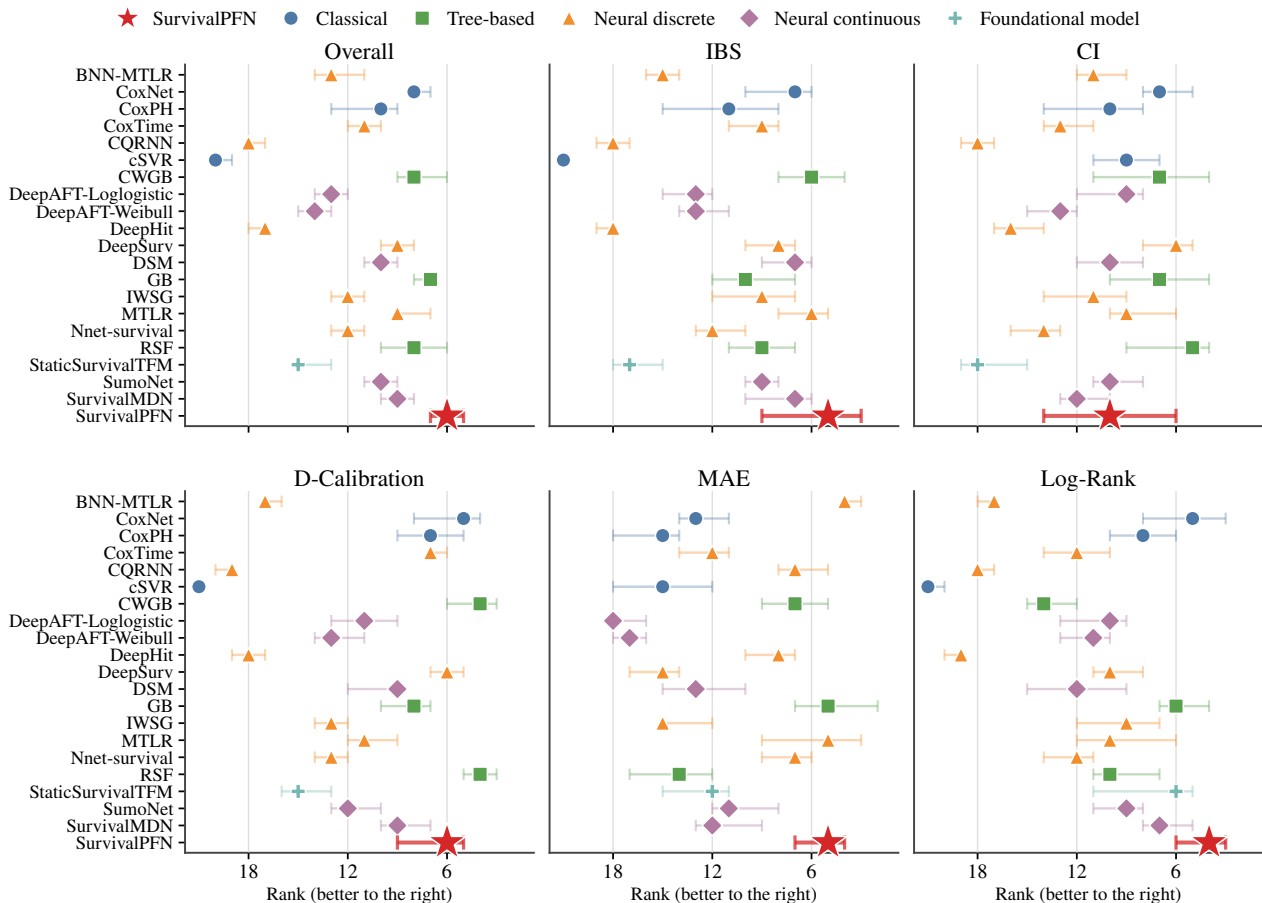

*Figure 5.* **Model ranks across 61 benchmark datasets.** Points/stars denote median ranks across datasets, with horizontal bars showing 95% bootstrap confidence intervals for the median rank.

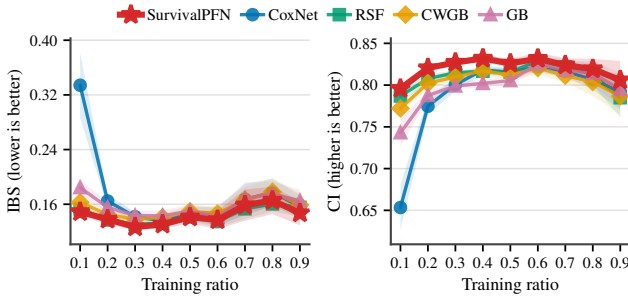

*Figure 6.* **Performance on the PBC dataset for SurvivalPFN and top-performing models.** Shaded regions denote standard errors over 10 repeated runs.

The comparison with general tabular foundation models further supports survival-specific pretraining. Off-the-shelf TFMs can be adapted by discarding censored observations or by expanding survival prediction into many time-indexed classification tasks, but both reductions lose some of the structure of censored data. SurvivalPFN instead treats $(X, T, \Delta)$ as the native observation and learns to return

the full event-time distribution directly. This appears especially important for probabilistic metrics such as IBS and D-calibration, where preserving uncertainty over the event time matters more than producing only a risk score.

## 5. Conclusions, Limitations, and Future Work

SurvivalPFN amortizes Bayesian survival prediction from right-censored data by pretraining on diverse identifiable DGPs and performing downstream inference in context. Across 61 held-out datasets, it offers strong aggregate performance and favorable runtime, suggesting that PFN-style in-context learning is a promising foundation for survival.

The central limitation is the identifiability assumption: under dependent censoring, the event-time distribution cannot be recovered from observed data alone without additional assumptions. Extending the prior to identifiable dependent-censoring families, such as copula-based mechanisms, is a natural direction. SurvivalPFN also inherits the long-context scalability challenges of current PFN models; improving performance on larger tables remains important future work.

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

## A. Notation

Table 1 summarizes all the notation used throughout the paper. Following the convention, we use uppercase letters for random variables and lowercase letters for realizations.

*Table 1.* Summary of notation in the paper.

| Notation | Definition |
|---|---|
| $\mathcal{X}, \mathbb{R}_+$ | Covariate space and nonnegative time domain. |
| $i \in \{1, \dots, N\}$ | Index for an individual observation. |
| $d$ | Number of covariates/features. |
| $X \in \mathcal{X}$ | Covariates. |
| $E \in \mathbb{R}_+$ | Latent event time of interest. |
| $C \in \mathbb{R}_+$ | Latent censoring time. |
| $T = \min(E, C)$ | Observed follow-up time. |
| $\Delta = \mathbb{1}\{E \leq C\}$ | Event indicator: $\Delta = 1$ for observed event and $\Delta = 0$ for right-censored. |
| $(x_i, t_i, \delta_i)$ | Observed right-censored tuple. |
| $(e_i, c_i)$ | Latent event and censoring times. |
| $\mathscr{D}$ | Random observed dataset. |
| $\mathcal{D} = \{(x_i, t_i, \delta_i)\}_{i=1}^N$ | Realization of $\mathscr{D}$. |
| $\mathcal{D}_\theta^{tr}, \mathcal{D}_\theta^{te}$ | Synthetic context/training set and query/test set sampled from DGP parameter $\theta$ during prior-data pretraining. |
| $\omega$ | Trainable parameters of SurvivalPFN. |
| $\theta \in \Theta$ | Latent parameter indexing a survival data-generating process. |
| $\theta^*$ | True DGP parameter for a downstream dataset. |
| $\pi(\cdot)$ | Prior distribution over survival DGPs used to generate synthetic training tasks. |
| $P^\theta$ | Joint law of $(X, E, C, T, \Delta)$ under DGP parameter $\theta$. |
| $P_{\text{obs}}^\theta$ | Observational law of $(X, T, \Delta)$ induced by $P^\theta$. |
| $f_{E\|X}^\theta(t \mid x)$ | Conditional density function for event time, under $\theta$. |
| $F_{E\|X}^\theta(t \mid x)$ | Conditional CDF for event time, $P^\theta(E \leq t \mid X = x)$. |
| $S_{E\|X}^\theta(t \mid x)$ | Conditional survival function for event time, $P^\theta(E > t \mid X = x) = 1 - F_{E\|X}^\theta(t \mid x)$. |
| $\lambda_{E\|X}^\theta(t \mid x)$ | Conditional hazard function, $\lambda_E^\theta(t \mid x) = f_{E\|X}^\theta(t \mid x)/S_{E\|X}^\theta(t \mid x)$. |
| $S_{C\|X}^\theta(t \mid x)$ | Conditional survival function for censoring time, $P^\theta(C \geq t \mid X = x)$. |
| $\lambda_{C\|X}^\theta(t \mid x)$ | Conditional censoring hazard. |
| $f_{E\|X,\mathscr{D}}(t \mid x, \mathcal{D})$ | Posterior predictive event-time distribution. |
| $S_{E\|X,\mathscr{D}}(t \mid x, \mathcal{D})$ | Posterior predictive survival distribution (PPSD). |
| $x^*$ | Query covariates for a new individual. |
| $\widetilde{\delta}^*$ | Query indicator supplied to SurvivalPFN. |
| $q_\omega(\cdot \mid x^*, \widetilde{\delta}^*, \mathcal{D})$ | SurvivalPFN's predictive distribution over discretized time bins. |
| $L$ | Number of time bins used to represent the predictive distribution. |
| $\{\mathcal{I}_\ell\}_{\ell=1}^L$ | SurvivalPFN's transformed-time bins used to represent the discretized predictive distribution. |
| $\mathcal{G} = \{t_1 < \dots < t_m\}$ | Discrete time grid for benchmarking discrete-time survival models. |
| $\text{CV}(T)$ | Coefficient of variation of observed times: $\text{CV}(T) = \text{sd}(T)/\mathbb{E}[T]$, when used in DGP diagnostics. |

## B. Identifiability and Non-identifiability under Right Censoring

In the context of survival analysis and causal inference, identifiability is the prerequisite for learning. If a model is non-identifiable, infinite data cannot distinguish between multiple underlying "truths" (*e.g.*, whether a drug works or if patients are simply dropping out due to side effects). Without identifiability, no consistent estimator exists, and any conclusions drawn from the data rely entirely on untestable assumptions rather than empirical evidence.

Tsiatis (1975, Theorem 2) first proved that the latent joint distribution of event time $E$ and censoring time $C$ is not identifiable from the (infinite) observed data $(T, \Delta)$ without additional assumptions. Formally, we restate the theorem using the notation that is consistent within this paper, by considering $E$ and $C$ as two competing events:

**Theorem B.1** (Non-identifiability; Marginal). *Let $S_{E,C}(e,c) = P(E > e, C > c)$ be an arbitrary joint survival function where $E$ and $C$ are dependent. There exists a different joint survival function $S_{E,C}^*(e,c)$, constructed such that $E$ and $C$ are independent $- S_{E,C}^*(e,c) = S_E^*(e)\, S_C^*(c) -$ which generates the exact same observed data distribution $P(T, \Delta)$.*

This means, without the assumption of random censoring ($E \perp C$), the marginal survival function $S_{E|X}(t)$ cannot be uniquely determined. This theorem can be easily extend to the conditional setting:

**Theorem B.2** (Non-identifiability; Conditional). *Let $X$ be a set of covariates. Let the true conditional joint survival function be $S_{E,C|X}(e,c \mid x) = P(E > e, C > c \mid X = x)$, where $E \not\perp C \mid X$. For any such dependent model, there exists a valid conditional independent model $S_{E,C|X}^*(e,c \mid x) = S_{E|X}^*(e \mid x) S_{C|X}^*(c \mid x)$ such that the observable distributions of $(T, \Delta \mid X)$ are identical.*

*Proof.* While the proof for this conditional non-identifiability is straightforward via following Tsiatis (1975)'s step, we present the proof for completeness.

Let the observed data be characterized by the conditional sub-survival functions:

$$
\begin{aligned}
\Phi(t \mid x) &= P(T > t, \Delta = 1 \mid x) \\
\Psi(t \mid x) &= P(T > t, \Delta = 0 \mid x)
\end{aligned}
$$

These functions completely describe the likelihood of the observed data.

We construct a "proxy" independent world, denoted by a superscript $^{\text{ind}}$, by defining its conditional hazards to match the observed cause-specific hazards of the original world:

$$
\begin{aligned}
\lambda_{E|X}^{\text{ind}}(t \mid x) &= \lim_{dt \to 0} \frac{\Pr^{\text{ind}}(t \leq T < t + dt, \Delta = 1 \mid T \geq t, X = x)}{dt} = \frac{-\frac{\partial}{\partial t}\Phi(t \mid x)}{\Phi(t \mid x) + \Psi(t \mid x)} \\
\lambda_{C|X}^{\text{ind}}(t \mid x) &= \lim_{dt \to 0} \frac{\Pr^{\text{ind}}(t \leq T < t + dt, \Delta = 0 \mid T \geq t, X = x)}{dt} = \frac{-\frac{\partial}{\partial t}\Psi(t \mid x)}{\Phi(t \mid x) + \Psi(t \mid x)}
\end{aligned}
$$

We define the marginal survival functions in the proxy world as:

$$
\begin{aligned}
S_{E|X}^{\text{ind}}(t \mid x) &= \exp\left( -\int_0^t \lambda_{E|X}^{\text{ind}}(u \mid x) du \right), \\
S_{C|X}^{\text{ind}}(t \mid x) &= \exp\left( -\int_0^t \lambda_{C|X}^{\text{ind}}(u \mid x) du \right).
\end{aligned}
$$

And the joint distribution (with conditional independence):

$$
S_{E,C|X}^{\text{ind}}(e,c \mid x) = S_{E|X}^{\text{ind}}(e \mid x) S_{C|X}^{\text{ind}}(c \mid x)
$$

To verify that this proxy world generates the same data $\Phi(t \mid x)$, we calculate the probability of observing an event in the proxy world:

$$
\begin{aligned}
\Phi^{\text{ind}}(t \mid x) &= \int_t^\infty f_{E|X}^{\text{ind}}(u \mid x) S_{C|X}^{\text{ind}}(u \mid x) du \\
&= \int_t^\infty \lambda_{E|X}^{\text{ind}}(u \mid x) S_{E|X}^{\text{ind}}(u \mid x) S_{C|X}^{\text{ind}}(u \mid x) du \\
&= \int_t^\infty \lambda_{E|X}^{\text{ind}}(u \mid x) S^{\text{ind}}(u, u \mid x) du
\end{aligned}
$$

Substituting the definitions of $\lambda_{E|X}^{\text{ind}}$ and noting that $S^{\text{ind}}(u, u \mid x) = \Phi(u \mid x) + \Psi(u \mid x)$ (the overall survival probability matches the sum of sub-survival functions):

$$
\begin{aligned}
\Phi^{\text{ind}}(t \mid x) &= \int_t^\infty \left( \frac{-\frac{\partial}{\partial u} \Phi(u \mid x)}{\Phi(u \mid x) + \Psi(u \mid x)} \right) (\Phi(u \mid x) + \Psi(u \mid x)) du \\
&= \int_t^\infty -\frac{\partial}{\partial u} \Phi(u \mid x) du \\
&= \Phi(t \mid x)
\end{aligned}
$$

Since $\Phi^{\text{ind}}(t \mid x) = \Phi(t \mid x)$ (and by symmetry $\Psi^{\text{ind}}(t \mid x) = \Psi(t \mid x)$), the independent model $S^{\text{ind}}$ is indistinguishable from the true dependent model $S$. □

While Theorem B.2 states that we cannot distinguish *dependent* from *independent* censoring using data alone, the corollary below establishes that if we are willing to assume independence (either marginal or conditional), the latent event distribution becomes identifiable.

**Corollary B.1** (Identifiability under Independence). *Suppose we restrict our attention to the class of models that satisfy conditional independent censoring, $E \perp C \mid X$ (including $E \perp C$). Then, the marginal survival function of the event, $S_{E|X}(t \mid x)$, is uniquely identifiable from the observed data distribution. That is, if two models $S$ and $S^{\text{ind}}$ both satisfy conditional independence but have different event marginals ($S_{E|X} \neq S_{E|X}^{\text{ind}}$), they must generate distinct observed data distributions.*

*Proof.* We prove this by contradiction. Let $\lambda_{E|X}(t \mid x)$ denote the *cause-specific* hazard derived purely from the data $(X, T, \Delta)$:

$$
\lambda_{E|X}(t \mid x) = \lim_{dt \to 0} \frac{P(t \leq T < t + dt, \Delta = 1 \mid T \geq t, X = x)}{dt}
$$

Let $\lambda(t \mid x)$ denote the *net* hazard of the event of interest:

$$
\lambda(t \mid x) = \lim_{dt \to 0} \frac{P(t \leq E < t + dt \mid E \geq t, X = x)}{dt}
$$

Under the assumption of conditional independence ($E \perp C \mid X$), standard survival theory dictates that the net hazard is equal to the cause-specific hazard: $\lambda_{E|X}(t \mid x) = \lambda(t \mid x)$. Since the hazard function uniquely defines the survival function via $S_{E|X}(t \mid x) = \exp(-\int_0^t \lambda_{E|X}(u \mid x) du)$, the latent distribution $S_{E|X}$ is uniquely determined by the observed function $\lambda_{E|X}$.

Now, consider two models with different marginals, $S_{E|X}^{(1)}(t \mid x) \neq S_{E|X}^{(2)}(t \mid x)$. This inequality implies their net hazards must differ: $\lambda^{(1)}(t \mid x) \neq \lambda^{(2)}(t \mid x)$. By the equality derived above, their observed cause-specific hazards must also differ: $\lambda_{E|X}^{(1)}(t \mid x) \neq \lambda_{E|X}^{(2)}(t \mid x)$. Different hazards imply different observed data distributions. Thus, the model is identifiable. □

These properties imply that while SurvivalPFN can be robustly trained under assumptions of marginal or conditional independence, it cannot learn dependent censoring mechanisms from observed data alone.

## C. Posterior-Predictive Consistency

This appendix formalizes Proposition C.1. The result concerns the Bayesian posterior predictive survival distribution (PPSD) defined in Equation C.1. It shows that, under an identifiable survival prior, the Bayesian PPSD is asymptotically consistent for the true conditional event-time survival function. We then state the corresponding idealized implication for SurvivalPFN when the transformer exactly amortizes this Bayesian target.

### C.1. Notation and regularity assumptions

Recall that a survival data-generating process (DGP) $P^\theta(X, E, C, T, \Delta)$ is indexed by $\theta \in \Theta$. We write $P^\theta_{\text{obs}}$ for the marginal distribution over the observable random variables $(X, T, \Delta)$. We still use $\pi(\cdot)$ to denote the prior over $\Theta$ induced by the synthetic prior-data generator used to pretrain SurvivalPFN.

For each $\theta$, let

$$S_{E|X,\Theta}(t \mid x, \theta) := \Pr(E > t \mid X = x, \Theta = \theta)$$

denote the conditional event-time survival function. For an observed dataset $\mathcal{D} = \{(X_i, T_i, \Delta_i)\}_{i=1}^N$, where $(X_i, T_i, \Delta_i) \overset{\text{i.i.d.}}{\sim} P^\theta_{\text{obs}}$, and a query covariate vector $x^*$, the Bayesian PPSD is (repeated from Equation C.1)

$$S_{E|X,\mathscr{D}}(t \mid x^*, \mathcal{D}) = \int_\Theta S_{E|X,\Theta}(t \mid x^*, \vartheta) f_{\Theta|\mathscr{D}}(\vartheta \mid \mathcal{D}) \, d\vartheta. \tag{C.1}$$

**Assumption C.1** (Regularity)**.** *We assume the following standard regularity conditions.*

1. *$(\Theta, \mathcal{B}_\Theta)$ is a standard Borel parameter space.*

2. *The maps $\theta \mapsto P^\theta$ and $\theta \mapsto P^\theta_{\text{obs}}$ are measurable.*

3. *The image set $\{P^\theta_{\text{obs}} : \theta \in \Theta\}$ is a Borel subset of the space of probability measures over $(X, T, \Delta)$.*

4. *For each time $t$ in the evaluation region, there exists a version of $S_{E|X,\Theta}(t \mid x^*, \theta)$ that is jointly measurable in $(x^*, \theta)$.*

These assumptions are technical rather than substantive. They ensure that priors, posteriors, conditional expectations, and the quotient construction below are well-defined. They are satisfied by the usual finite-dimensional parameter spaces and measurable simulators used in statistical and machine learning models.

### C.2. Observed-law equivalence and survival identifiability

The full latent law $P^\theta(X, E, C, T, \Delta)$ is generally not identifiable from right-censored observations. The data can identify only the observed law $P^\theta_{\text{obs}}$ over $(X, T, \Delta)$. We therefore group DGP parameters by observational equivalence:

$$\theta_1 \sim \theta_2 \iff P^{\theta_1}_{\text{obs}} = P^{\theta_2}_{\text{obs}}.$$

Let

$$\mathcal{Q} := \Theta/\sim$$

denotes the set of observational quotient space (equivalence classes) induced by $\sim$, and let $[\theta] \in \mathcal{Q}$ denote the equivalence class of $\theta$. Each class $[\theta]$ contains all latent survival DGPs that are indistinguishable.

**Definition C.1** (Survival-identifiable prior)**.** *Fix a time $t$ in the evaluation region. We say that the prior $\pi$ is survival-identifiable at time $t$ if there exists a measurable map*

$$F_t : \mathcal{X} \times \mathcal{Q} \to [0, 1]$$

*such that, for $\pi$-almost every $\theta$ and $P^\pi_X$-almost every $x$ (except where $\theta$ or $x$ has probability 0),*[1]

$$S_{E|X,\Theta}(t \mid x, \theta) = F_t(x, [\theta]).$$

*Equivalently, within the support of the prior, any two DGPs that induce the same observational distribution over $(X, T, \Delta)$ must also induce the same conditional event-time survival function at time $t$.*

The conditional independent censoring and positivity assumptions discussed in Section 2 provide sufficient conditions for this definition: under those assumptions, $S_{E|X}(t \mid x)$ is a functional of the observed law $P(X, T, \Delta)$ on the identifiable time region.

---

[1]In the following, we will just use the phrase "every $\theta$" and "every $x$" for simplicity.

## C.3. Consistency of the Bayesian PPSD

**Proposition C.1** (Formal consistency). *Fix a time $t$ in the evaluation region. Under Assumption C.1, there exist sets $\mathcal{X}_0 \subseteq \mathcal{X}$ and $\Theta_0 \subseteq \Theta$ with*

$$P_X^\pi(\mathcal{X}_0) = 1, \qquad \pi(\Theta_0) = 1,$$

*such that for every $x^* \in \mathcal{X}_0$ and every $\theta^* \in \Theta_0$, then*

$$S_{E|X,\mathscr{D}}(t \mid x^*, \mathcal{D}) \xrightarrow[N\to\infty]{\text{a.s.}} S_{E|X,\Theta}(t \mid x^*, \theta^*) \tag{C.2}$$

*if and only if the prior $\pi$ is survival-identifiable at time $t$.*

*For any finite or countable evaluation grid $\mathcal{G}$, the same result holds simultaneously for all $t \in \mathcal{G}$ by intersecting the corresponding full-measure sets.*

*Proof.* We first define the quotient-level target. For fixed $t$ and $x$, we define

$$M_t(x^*, [\theta]) := \mathbb{E}_\pi \left[ S_{E|X,\Theta}(t \mid x^*, \vartheta) \mid [\vartheta] = [\theta] \right].$$

This is the prior-average survival probability among all DGPs that induce the same observed law as $\theta$. Since $0 \leq S_{E|X,\Theta}(t \mid x^*, \vartheta) \leq 1$, this conditional expectation is integrable.

By construction, two different elements of $\mathcal{Q}$ correspond to two different observational distributions. Thus, the quotient parameter $[\theta]$ is identifiable from the observed distribution. Assumption C.1 ensures that the quotient model is measurable and that Doob's consistency theorem (Doob, 1949) applies to posterior expectations of integrable functions on this quotient space.

Therefore, for every true parameter $\theta^*$ (except where $\theta$ has probability 0), if $\mathcal{D} \sim P_{\text{obs}}^{\theta^*}$, then

$$\mathbb{E}_\pi \left[ M_t(x^*, [\theta]) \mid \mathcal{D} \right] \xrightarrow[N\to\infty]{\text{a.s.}} M_t(x^*, [\theta^*]). \tag{C.3}$$

We now show that the left-hand side of Equation C.3 is the Bayesian PPSD. Since the observed data distribution depends on $\theta$ only through the equivalence class $[\theta]$, we have the conditional independence

$$\mathcal{D} \perp \theta \mid [\theta].$$

Hence,

$$\begin{aligned}
\mathbb{E}_\pi \left[ M_t(x^*, [\theta]) \mid \mathcal{D} \right] &= \mathbb{E}_\pi \left[ \mathbb{E}_\pi \left[ S_{E|X,\Theta}(t \mid x^*, \theta) \mid [\theta] \right] \mid \mathcal{D} \right] \\
&= \mathbb{E}_\pi \left[ \mathbb{E}_\pi \left[ S_{E|X,\Theta}(t \mid x^*, \theta) \mid [\theta], \mathcal{D} \right] \mid \mathcal{D} \right] \\
&= \mathbb{E}_\pi \left[ S_{E|X,\Theta}(t \mid x^*, \theta) \mid \mathcal{D} \right] \\
&= \int_\Theta S_{E|X,\Theta}(t \mid x^*, \vartheta) f_{\Theta|\mathscr{D}}(\vartheta \mid \mathcal{D}) \, d\vartheta \\
&= S_{E|X,\mathscr{D}}(t \mid x^*, \mathcal{D}).
\end{aligned}$$

Combining this equality with Equation C.3 gives

$$S_{E|X,\mathscr{D}}(t \mid x^*, \mathcal{D}) \xrightarrow[N\to\infty]{\text{a.s.}} M_t(x^*, [\theta^*]). \tag{C.4}$$

Without identifiability, this is the strongest possible conclusion: the PPSD converges to the prior-average survival function over DGPs that are observationally equivalent to the truth.

Finally, note $\pi$ is survival-identifiable at time $t$. Then, by Definition C.1, all DGPs in the same observational equivalence class have the same survival probability at $(t, x^*)$. Therefore,

$$\begin{aligned}
M_t(x^*, [\theta^*]) &= \mathbb{E}_\pi \left[ S_{E|X,\Theta}(t \mid x^*, \vartheta) \mid [\vartheta] = [\theta^*] \right] \\
&= S_{E|X,\Theta}(t \mid x^*, \theta^*).
\end{aligned}$$

Substituting this into Equation C.4 proves Equation C.2. This establishes sufficiency.

For necessity, suppose that the Bayesian PPSD is consistent in the sense of Equation C.2 for every $\theta^*$. From Equation C.4, the same sequence also converges almost surely to $M_t(x^*, [\theta^*])$. By uniqueness of almost-sure limits,

$$S_{E|X,\Theta}(t \mid x^*, \theta^*) \;=\; M_t(x^*, [\theta^*])$$

for every $\theta^*$. Since the right-hand side depends on $\theta^*$ only through its observational equivalence class $[\theta^*]$, the survival functional $S_{E|X,\Theta}(t \mid x^*, \theta^*)$ also depends only on $[\theta^*]$. Thus, Definition C.1 holds with $F_t(x^*, [\theta]) = M_t(x^*, [\theta])$, up to arbitrary definition on null sets. Hence the prior is survival-identifiable at time $t$. $\qquad\square$

## C.4. Implication for SurvivalPFN

Proposition C.1 characterizes the Bayesian posterior predictive target. SurvivalPFN is a finite neural network trained to approximate this target, so the theorem does not by itself prove finite-sample or finite-capacity consistency of the trained transformer. It does, however, imply consistency in the idealized exact-amortization limit.

**Corollary C.1** (Consistency of SurvivalPFN)**.** *Assume the conditions of Proposition C.1. Suppose that, for $\widetilde{\delta}^* = 1$, SurvivalPFN exactly amortizes the Bayesian posterior predictive event-time distribution, so that its predicted survival curve satisfies*

$$\widehat{S}_\omega(t \mid x^*, \mathcal{D}) \;=\; S_{E|X,\mathscr{D}}(t \mid x^*, \mathcal{D}).$$

*Then, for every true DGP parameter $\theta^*$ and every $x$,*

$$\widehat{S}_\omega(t \mid x^*, \mathcal{D}) \xrightarrow[N\to\infty]{\text{a.s.}} S_{E|X,\Theta}(t \mid x^*, \theta^*).$$

*More generally, the same conclusion holds if the amortization error vanishes:*

$$\left|\widehat{S}_\omega(t \mid x^*, \mathcal{D}) - S_{E|X,\mathscr{D}}(t \mid x^*, \mathcal{D})\right| \xrightarrow[N\to\infty]{} 0.$$

*Proof.* The exact-amortization case follows immediately by substituting $\widehat{S}_\omega(t \mid x^*, \mathcal{D}) = S_{E|X,\mathscr{D}}(t \mid x^*, \mathcal{D})$ into Proposition C.1. The vanishing-error case follows from the triangle inequality:

$$\left|\widehat{S}_\omega(t \mid x^*, \mathcal{D}) - S_{E|X,\Theta}(t \mid x^*, \theta^*)\right|$$

$$\leq \left|\widehat{S}_\omega(t \mid x^*, \mathcal{D}) - S_{E|X,\mathscr{D}}(t \mid x^*, \mathcal{D})\right| + \left|S_{E|X,\mathscr{D}}(t \mid x^*, \mathcal{D}) - S_{E|X,\Theta}(t \mid x^*, \theta^*)\right|.$$

The first term vanishes by assumption, and the second term vanishes almost surely by Proposition C.1. $\qquad\square$

# D. Additional SurvivalPFN Details

## D.1. Synthetic Prior and DGP Simulation

SurvivalPFN is pretrained on synthetic right-censored survival datasets sampled from a prior $\pi(\cdot)$ over identifiable survival data-generating processes. A draw $\theta \sim \pi(\cdot)$ specifies all random choices needed to generate one synthetic task, including the covariate distribution, event-time mechanism, censoring-time mechanism, censoring type, and target censoring rate.

The simulator returns both the observed survival dataset

$$\mathcal{D}_\theta^{tr} \;=\; \{(x_i, t_i, \delta_i)_\theta\}_{i=1}^N,$$

and the corresponding latent event and censoring times $\{(e_i, c_i)_\theta\}_{i=1}^N$, which are used only for constructing supervised query targets during prior-data training.

**Random Tabular Generators.**   All survival priors are built from a generic tabular generator. We write $G$ for a random table generator that can sample either an unconditional table or a conditional table. Concretely, a call to $G$ first samples generator-specific latent parameters $\zeta$, and then produces either

$$X \sim G_\zeta(\cdot), \qquad Y \mid X \sim G_\zeta(\cdot \mid X).$$

based on the requirement.

This abstraction also lets the same survival-prior design use different tabular generators, including PFN-style random multilayer perceptrons, random structural causal models, tree-based generators, or mixtures of these sources.

**Censoring Mechanisms.** We include several censoring mechanisms to cover common right-censoring regimes while retaining identifiability.

*Uniform censoring.* Event times are generated from the event-time prior, while censoring times are sampled from a uniform distribution. Depending on the time-generation family, the support is either based on the generated event-time range or on a sampled global time horizon:

$$C_i \sim \mathrm{Unif}\left(\min_j E_j, \max_j E_j\right), \qquad \text{or} \qquad C_i \sim \mathrm{Unif}(0, t_{\max}).$$

*Random censoring.* Event times are generated conditionally on $X$, while censoring times are generated from a separate unconditional tabular mechanism:

$$E \mid X \sim \Pr(E \mid X), \qquad C \sim \Pr(C).$$

This produces censoring distributions that are more diverse than simple uniform censoring while remaining independent of the event-time noise conditional on the task.

*Administrative censoring.* Administrative censoring simulates staggered study entry with a common study end date. The simulator samples entry times $A_i$, chooses a fixed administrative end time $a_*$, and sets

$$C_i = a_* - A_i.$$

This captures a common survival-analysis setting in which subjects enter at different calendar times but are all censored at the same study termination date.

*Independent censoring.* For independent (or covariate-dependent) censoring, both event and censoring times are generated conditional on $X$, but from independent blocks:

$$E \perp C \mid X, \qquad \Pr(E, C \mid X) = \Pr(E \mid X) \Pr(C \mid X).$$

This mechanism is central to the identifiable-prior construction: it allows censoring to depend on covariates while preserving the standard conditional independent censoring assumption used for nonparametric identification.

**Prior Family 1: Naive Survival Prior.** The simplest prior treats generated table values as raw event and censoring times. Let $G_\zeta$ denote a conditional event/censoring table generator. For the event-time branch, we sample

$$E = G_\zeta(X; 1),$$

where $G_\zeta(X; 1)$ denotes one conditional output column given $X$. The censoring branch is then chosen according to the censoring type: uniform censoring samples $C$ from a uniform distribution; tabular censoring samples $C$ from an unconditional table generator; administrative censoring constructs $C$ from entry times; and the conditional independent branch samples separate event and censoring columns conditional on $X$. This prior is intentionally simple and flexible: it exposes the model to irregular, nonparametric time patterns without imposing an explicit survival-time family.

**Prior Family 2: Survival-Distribution Prior.** The second prior generates smooth distributions using random monotone maps. For each dataset, the simulator samples a time horizon $t_{\max} > 0$, a number of knots $K$, and unconstrained coefficients

$$c = (c_1, \ldots, c_K) \in \mathbb{R}^K.$$

These coefficients are converted into positive increments

$$\Delta_j = \frac{\exp(c_j)}{\sum_{\ell=1}^{K} \exp(c_\ell)}, \qquad j = 1, \ldots, K,$$

which define monotone control points

$$b_0 = 0, \qquad b_j = \sum_{\ell=1}^{j} \Delta_\ell, \qquad j = 1, \ldots, K.$$

The resulting monotone Bernstein map is

$$f_c(u) = \sum_{j=0}^{K} b_j \binom{K}{j} u^j (1-u)^{K-j}, \qquad u \in [0,1].$$

A raw time is then sampled by pushing uniform noise through this monotone map:

$$\tau = t_{\max} f_c(U), \qquad U \sim \text{Unif}(0,1).$$

This construction can be viewed as a random smooth quantile-like function. It gives a flexible family of event-time and censoring-time distributions without committing to a fixed parametric hazard form. Under conditional independent censoring, event and censoring coefficient blocks are generated independently given $X$:

$$E_i = t_{\max} f_{c_i^E}(U_i^E),$$
$$C_i = t_{\max} f_{c_i^C}(U_i^C),$$

with independent $U_i^E, U_i^C \sim \text{Unif}(0,1)$ and independent coefficient blocks conditional on $x_i$.

**Prior Family 3: Mixture Prior.** The third prior samples event and censoring times from mixtures of distributions. In our implementation, the component family is sampled from Weibull and lognormal distributions, and the number of components $K_{\text{mix}}$ is sampled from a finite candidate set. For each row $i$, a table generator emits hidden values

$$h_i = (a_{i1}, b_{i1}, r_{i1}, \ldots, a_{iK_{\text{mix}}}, b_{iK_{\text{mix}}}, r_{iK_{\text{mix}}}).$$

The logits $r_{ij}$ define mixture weights

$$\pi_{ij} = \frac{\exp(r_{ij})}{\sum_{\ell=1}^{K_{\text{mix}}} \exp(r_{i\ell})}, \qquad \sum_{j=1}^{K_{\text{mix}}} \pi_{ij} = 1,$$

and a component index is sampled as

$$Z_i \sim \text{Categorical}(\pi_{i1}, \ldots, \pi_{iK_{\text{mix}}}).$$

For a Weibull component, positive shape and scale parameters are obtained by

$$\kappa_{ij} = \text{softplus}(a_{ij}) + 0.1,$$
$$\lambda_{ij} = \text{softplus}(b_{ij}) + 0.1,$$

and the sampled time is

$$\tau_i = \lambda_{iZ_i} [-\log(1 - U_i)]^{1/\kappa_{iZ_i}}, \qquad U_i \sim \text{Unif}(0,1).$$

For a lognormal component,

$$\mu_{ij} = \text{softplus}(a_{ij}),$$
$$\sigma_{ij} = \text{softplus}(b_{ij}),$$

and

$$\log \tau_i = \mu_{iZ_i} + \sigma_{iZ_i} \varepsilon_i, \qquad \varepsilon_i \sim \mathcal{N}(0,1).$$

This mixture prior provides a complementary source of smooth positive-time distributions with heavy tails and multimodality.

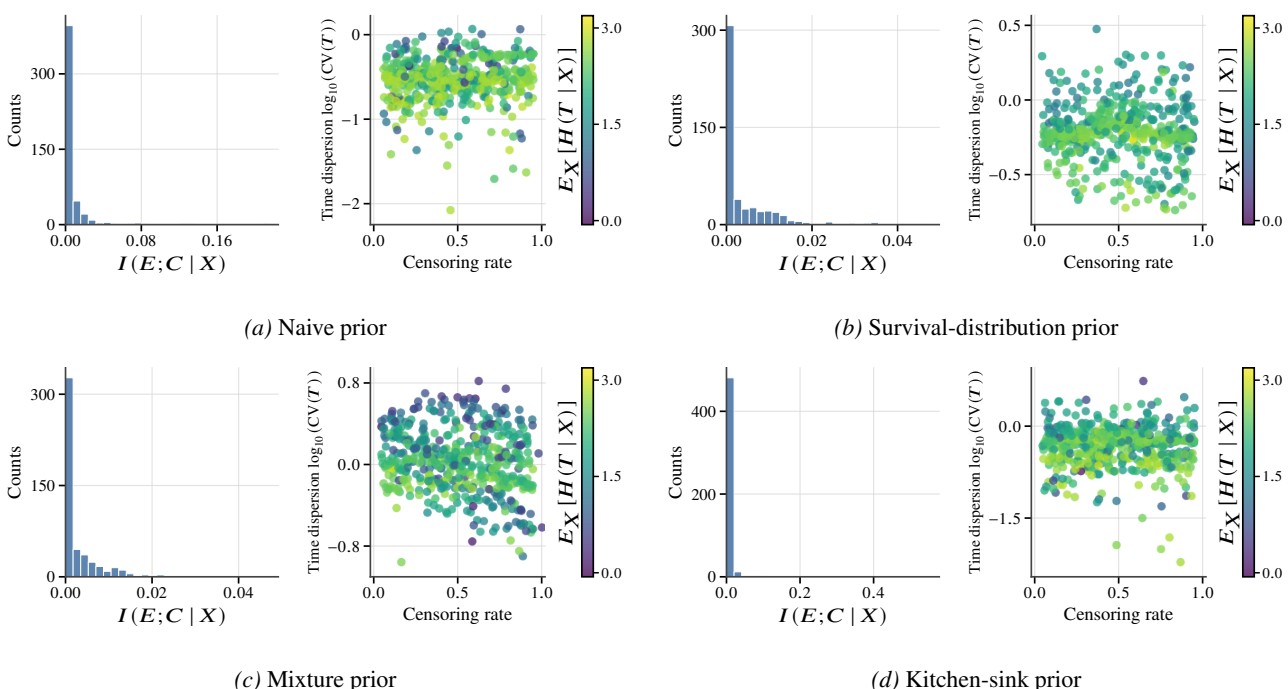

*(a)* Naive prior

*(b)* Survival-distribution prior

*(c)* Mixture prior

*(d)* Kitchen-sink prior

*Figure 7.* Prior-quality diagnostics across four synthetic prior families, with 500 sampled datasets per family. Each panel summarizes the induced dependence between event and censoring times, censoring-rate coverage, time-scale dispersion, and conditional event-time uncertainty.

**Prior Family 4: Kitchen-Sink Meta Prior.** Finally, SurvivalPFN can use a meta-prior that mixes multiple complete survival priors. Let $P_1, \ldots, P_M$ denote child prior generators and let $w_1, \ldots, w_M$ be nonnegative mixture weights with $\sum_m w_m = 1$. The kitchen-sink prior samples

$$M^* \sim \text{Categorical}(w_1, \ldots, w_M), \qquad \mathcal{D}_\theta^{tr} \sim P_{M^*}.$$

Equivalently, the induced prior over synthetic datasets is

$$P_{\text{sink}}(\mathcal{D}) = \sum_{m=1}^{M} w_m P_m(\mathcal{D}).$$

This mixture increases prior diversity by combining direct table-output mechanisms, smooth monotone distributional mechanisms, and positive-time mixture mechanisms. In the current default configuration, the kitchen-sink prior places most mass on the direct table-output and monotone survival-distribution priors, while the exact mixture weights remain configurable.

**Diagnostics for Synthetic DGPs.** We further examine the synthetic DGPs induced by each prior family in Figures 7 and 8. These diagnostics are designed to check the two desiderata of the prior: it should remain close to the identifiable independent-censoring regime, while still covering a broad range of survival-data regimes. For each prior family, we sample 500 synthetic datasets, each with 1,024 observations, and compute dataset-level and curve-level summaries.

Figure 7 reports scalar diagnostics for each generated dataset. The left subpanel estimates the conditional mutual information $I(E; C \mid X)$ between the latent event time $E$ and censoring time $C$ after conditioning on covariates $X$. Since our simulator is constructed under conditional independence, values concentrated near zero provide an empirical sanity check that the generated censoring process does not introduce substantial residual event-censoring dependence beyond $X$. The right subpanel summarizes dataset diversity: each point is one generated dataset, with the x-axis showing the censoring rate, the y-axis showing the observed-time dispersion $\log_{10}(\text{CV}(T))$, where $\text{CV}(T) = \text{std}(T)/|\text{mean}(T)|$ is the coefficient of variation of the observed times, and the color showing the conditional observed-time entropy $\mathbb{E}_X[H(T \mid X)]$. Thus,

this panel visualizes how broadly each prior covers censoring rates, relative time-scale variation, and residual outcome uncertainty after conditioning on covariates.

Several trends are apparent from these plots. First, across all four prior families, the estimated conditional mutual information $I(E; C \mid X)$ is highly concentrated near zero, suggesting that the generated datasets largely remain within the intended conditional independent censoring regime. This provides an empirical sanity check that the prior does not typically generate strongly informative censoring structures. Second, the scatter plots show that the priors cover a broad range of right-censored survival regimes. The generated datasets span nearly the full range of censoring rates, from lightly censored to heavily censored settings. They also cover different observed-time dispersion regimes, and different levels of residual conditional uncertainty, as measured by $\mathbb{E}_X[H(T \mid X)]$.

Figure 8 complements these scalar summaries with curve-level diagnostics. For each generated dataset, we compute the empirical latent event-time survival curve $P(E > t)$, the empirical latent censoring-time survival curve $P(C > t)$, and the Kaplan-Meier estimate $\widehat{S}_{\mathrm{KM}}(t)$ from the observed pairs $(T, \Delta)$. The solid line denotes the pointwise median curve across generated datasets, while the dark and light bands denote the pointwise 25th-75th and 10th-90th percentile ranges, respectively. These curves show that the priors induce a wide range of event-time, censoring-time, and observed survival shapes, including different time scales, tail behaviors, and censoring-distorted Kaplan-Meier patterns. Together, Figures 7 and 8 show that the proposed prior families generate diverse survival tasks while maintaining the identifiable right-censoring structure required by our theoretical framework. These curve-level summaries further confirm that the diversity of DGPs. The empirical distributions vary substantially across generated datasets, with different decay rates, time scales, and tail behaviors.

### D.2. Architecture Details

SurvivalPFN uses the same PFN-style transformer architecture as TabDPT (Ma et al., 2024) and CausalPFN (Balazadeh et al., 2025), with only task-specific changes to the token contents and output interpretation. Each context row $(x_i, t_i, \delta_i) \in \mathcal{D}_\theta^{tr}$ is represented as a single token by combining embeddings of the covariates $x_i$, observed time $t_i$, and event indicator $\delta_i$. Each query row is represented by embeddings of the query covariate $x_\theta^*$ and the query indicator $\widetilde{\delta}^*$. We use linear embedding layers and omit positional encodings, so the context is treated as a set rather than an ordered sequence.

All context and query tokens are passed through a 20-layer transformer encoder with hidden dimension 384, RMS query-key normalization, and parallel SwiGLU feed-forward blocks. The attention mask follows the standard PFN design (Hollmann et al., 2022; 2025): context tokens attend to one another, while query tokens attend only to the context. This masking prevents information leakage across queries and allows the model to process many query instances in parallel. The output representation of each query token is projected to $L = 1024$ logits, and a softmax produces a discretized PPD over the transformed time bins:

$$q_\omega(\cdot \mid x_\theta^*, \widetilde{\delta}^*, \mathcal{D}_\theta^{tr}) \; = \; \left[ q_{\omega,\ell}(x_\theta^*, \widetilde{\delta}^*, \mathcal{D}_\theta^{tr}) \right]_{\ell=1}^{L}.$$

Depending on $\widetilde{\delta}^*$, this distribution is interpreted as an approximation to either the PPD for event time or the PPD for censoring time. The corresponding PPSD or PPCD is obtained by summing the predicted tail probability mass across bins.

The full model has approximately 20M parameters and is trained in two stages: (i) a predictive phase that follows standard predictive PFN training from Ma et al. (2024), and (ii) a survival phase that optimizes the survival likelihood or cross-entropy loss. We use AdamW (Kingma & Ba, 2014) with warmup and cosine annealing in the predictive phase, and switch to the schedule-free optimizer (Defazio et al., 2024) in the survival phase. The maximum context length is 16K in the first phase and 2,048 in the second. The predictive phase is trained on four A100 GPUs for up to one week, followed by two and half days of survival-phase training on one H100 GPU.

Training is parallelized over both synthetic context and query tokens. At each gradient step, we sample $B_\theta$ independent DGPs from the prior, generate one context table for each DGP, and draw $B_q$ query rows per datasets. These $B_\theta B_q$ query predictions are computed in a single batched forward pass, and the final loss is averaged over all DGP-query pairs. This batching strategy is identical in spirit to the parallel training procedure used in CausalPFN; see Algorithm 1 in Balazadeh et al. (2025) for the full procedure.

### D.3. Monotone Time Transformations

Survival times are nonnegative and often highly skewed. Directly discretizing raw time can therefore allocate too many bins to sparse tail regions or make the learned distribution sensitive to dataset-specific time scales. To address this, SurvivalPFN applies a context-fitted monotone transformation before discretization.

For each dataset $\mathcal{D} = \{(x_i, t_i, \delta_i)\}_{i=1}^{N}$, let $g : \mathbb{R}_+ \to \mathcal{Z}$ denote the dataset-specific transformation fitted on the observed times for the context tokens $\{t_i\}_{i=1}^{N_c}$, where $N_c$ is the size of context tokens. The observed times and query targets for context tokens are mapped into model space as

$$
\begin{aligned}
z_i &= g(t_i), \\
z_e^* &= g(e^*), \\
z_c^* &= g(c^*).
\end{aligned}
$$

The histogram likelihood Equation D.1 or cross-entropy loss in Equation D.2 is then evaluated in this transformed space.

To make prediction, model-space bin edges are mapped back to raw time through $g^{-1}$, so that the final PPSD/PPCD is reported on the original time scale. In all cases, $g$ is monotone increasing, preserving the temporal ordering of events and censoring times.

**`lognormal2normal` Transformation.** The `lognormal2normal` transformation uses a simple parametric normalization motivated by the positivity and right-skewness of survival times. For each dataset, we first calculate the mean $m$ and variance $s^2$ on the context tokens. We fit a lognormal distribution whose raw-time mean and standard deviation match $m$ and $s^2$. If $T = \exp(\mu + \sigma Z)$ with $Z \sim \mathcal{N}(0, 1)$, then

$$
\begin{aligned}
\sigma^2 &= \log\left(1 + \frac{s^2}{m^2}\right), \\
\mu &= \log(m) - \frac{1}{2}\sigma^2.
\end{aligned}
$$

The forward transformation maps a raw time $t > 0$ to its fitted normal coordinate:

$$
g_{\text{LN}}(t) = \frac{\log t - \mu}{\sigma}.
$$

The inverse transformation is

$$
g_{\text{LN}}^{-1}(z) = \exp(\mu + \sigma z).
$$

Thus, fixed-width bins in model space correspond to adaptive, nonuniform bins in raw time. This transformation is smooth, bijective on $\mathbb{R}_+$, preserves time ordering, and can extrapolate beyond the largest observed time. Its main limitation is that it imposes a lognormal shape on the observed-time distribution; this parametric normalization may not allocate resolution optimally when the parametric form is incorrect.

**`time2quantile` Transformation.** The `time2quantile` transformation is a nonparametric, context-adaptive alternative. It maps raw time into an empirical quantile coordinate in $[0, 1]$. For all the context tokens in the dataset, sort the observed times and collapse duplicates to obtain unique knots

$$
0 = a_0 < a_1 < \cdots < a_K = \max_i t_i.
$$

Each knot $a_j$ is assigned its right-continuous empirical CDF value

$$
q_j = \widehat{F}(a_j) = \frac{1}{N_c} \sum_{i=1}^{N_c} \mathbb{1}\{t_i \le a_j\}, \qquad q_0 = 0, \quad q_K = 1.
$$

The forward transformation is the piecewise-linear interpolation of these knots:

$$
g_{\text{Q}}(t) = q_j + \frac{t - a_j}{a_{j+1} - a_j}(q_{j+1} - q_j), \qquad a_j \le t \le a_{j+1}.
$$

Values above the largest time are mapped to $1$. The inverse transformation swaps the axes and linearly interpolates from quantile space back to raw time:

$$g_{\mathrm{Q}}^{-1}(q) \;=\; a_j + \frac{q - q_j}{q_{j+1} - q_j}(a_{j+1} - a_j), \qquad q_j \;\leq\; q \;\leq\; q_{j+1}.$$

For this transformation, the model-space range is fixed to $[0, 1]$. Uniform bins in quantile space become adaptive raw-time bins: regions with many observed times receive finer resolution, while sparse regions receive wider bins.

The `time2quantile` transformation makes no parametric assumption on the time distribution and is robust to changes in time units or monotone rescalings of raw time. However, because it is fitted from the empirical distribution of $\{t_i\}_{i=1}^{N_c}$, it is context-local and cannot resolve the tail shape beyond $\max_i t_i$; all larger times are mapped to quantile $1$, with probability mass beyond this point represented only by the final residual histogram bin.

**Summary.** The two transformations reflect complementary design choices. The `lognormal2normal` transformation provides a smooth positive-time coordinate with parametric tail extrapolation, while `time2quantile` provides a fully context-adaptive coordinate that normalizes all tasks to the common interval $[0, 1]$. Both transformations preserve time ordering and allow SurvivalPFN to allocate discretization resolution more effectively than raw-time binning.

### D.4. Training Objective

SurvivalPFN is trained to predict a discretized PPD over transformed time. For a query with indicator $\widetilde{\delta}^*$, define the latent supervised target

$$r_\theta^*(\widetilde{\delta}^*) \;=\; \widetilde{\delta}^* e_\theta^* + \left(1 - \widetilde{\delta}^*\right) c_\theta^*.$$

Thus, $\widetilde{\delta}^* = 1$ asks for the latent event time, whose posterior predictive tail gives the PPSD, while $\widetilde{\delta}^* = 0$ asks for the latent censoring time, corresponding to the PPCD.

Let $g_{\mathcal{D}_\theta^{tr}} : \mathbb{R}_+ \to \mathcal{Z}$ be the monotone transformation fitted from the observed context times in $\mathcal{D}_\theta^{tr}$, and let $\{\mathcal{I}_\ell\}_{\ell=1}^L$ denote the ordered bins in transformed-time space. Define the bin-index map

$$\kappa_{\mathcal{D}_\theta^{tr}}(r) \;=\; \ell \quad \text{if} \quad g_{\mathcal{D}_\theta^{tr}}(r) \in \mathcal{I}_\ell,$$

with boundary clipping handled by the same convention used in implementation. Equivalently, define the one-hot target

$$b_{\ell, \mathcal{D}_\theta^{tr}}(r) \;=\; \mathbb{I}\Big\{\kappa_{\mathcal{D}_\theta^{tr}}(r) = \ell\Big\}, \qquad \ell = 1, \ldots, L.$$

**Likelihood loss.** The main SurvivalPFN checkpoint is trained with the one-hot discrete negative log-likelihood over transformed-time bins:

$$\mathrm{NLL}(r_\theta^* \,\|\, q_\omega) \;=\; -\log q_{\omega, \kappa_{\mathcal{D}_\theta^{tr}}(r_\theta^*)}\Big(x_\theta^*, \widetilde{\delta}^*, \mathcal{D}_\theta^{tr}\Big).$$

Equivalently,

$$\mathrm{NLL}(r_\theta^* \,\|\, q_\omega) \;=\; -\sum_{\ell=1}^L b_{\ell, \mathcal{D}_\theta^{tr}}(r_\theta^*) \log q_{\omega, \ell}\Big(x_\theta^*, \widetilde{\delta}^*, \mathcal{D}_\theta^{tr}\Big).$$

The corresponding population objective is

$$\mathcal{L}_{\mathrm{NLL}}(\omega) \;=\; \mathbb{E}_{\theta \sim \pi(\cdot)} \, \mathbb{E}_{\mathcal{D}_\theta^{tr}, \, x_\theta^*, \, e_\theta^*, \, c_\theta^*, \, \widetilde{\delta}^*} \left[ -\log q_{\omega, \kappa_{\mathcal{D}_\theta^{tr}}(r_\theta^*(\widetilde{\delta}^*))}\Big(x_\theta^*, \widetilde{\delta}^*, \mathcal{D}_\theta^{tr}\Big) \right]. \tag{D.1}$$

This is the objective used by the best validation checkpoint reported in the main results.

**Smoothed cross-entropy variant.** We also consider a smoothed cross-entropy loss, which is used in TabDPT (Ma et al., 2024) and CausalPFN (Balazadeh et al., 2025). Instead of assigning all mass to one bin, the target time is converted into a narrow histogram in transformed-time space:

$$a_{\ell, \mathcal{D}_\theta^{tr}}^{(\sigma)}(r) \;=\; \int_{\mathcal{I}_\ell} \alpha_\sigma\Big(z; \, g_{\mathcal{D}_\theta^{tr}}(r)\Big) \, dz, \qquad \ell = 1, \ldots, L,$$

where $\alpha_\sigma$ is a narrow smoothing density, implemented as a Gaussian centered at $g_{\mathcal{D}_\theta^{tr}}(r)$ with a predefined variance. The smoothed cross-entropy loss is

$$\mathrm{SCE}_\sigma(r_\theta^* \| q_\omega) \;=\; -\sum_{\ell=1}^{L} a_{\ell, \mathcal{D}_\theta^{tr}}^{(\sigma)}(r_\theta^*) \log q_{\omega, \ell}\left(x_\theta^*, \widetilde{\delta}^*, \mathcal{D}_\theta^{tr}\right). \tag{D.2}$$

As $\sigma \to 0$, the smoothed target $a_{\ell, \mathcal{D}_\theta^{tr}}^{(\sigma)}(r)$ reduces to the one-hot target $b_{\ell, \mathcal{D}_\theta^{tr}}(r)$, and the smoothed cross-entropy recovers the discrete NLL. In our experiments, this loss is treated as an alternative training option and evaluated in the ablation study.

**Note.** This prior-data likelihood differs from the standard right-censored survival likelihood in Section 2. The standard observed-data likelihood trains on partially observed query outcomes $(t^*, \delta^*)$ and must account for censored queries through survival-tail probabilities. In contrast, SurvivalPFN training uses simulator-provided latent query times $e_\theta^*$ or $c_\theta^*$ as supervision.

### D.5. Inference Procedure

At inference time, SurvivalPFN takes an observed survival dataset $\mathcal{D}$ as context and one or more query covariates $x^*$. For survival prediction, we set the query indicator to $\widetilde{\delta}^* = 1$, so that the model outputs a discretized posterior predictive event-time distribution

$$q_\omega\left(\cdot \,\middle|\, x^*, \widetilde{\delta}^* = 1, \mathcal{D}\right) \;=\; \left[q_{\omega, \ell}(x^*, \widetilde{\delta}^* = 1, \mathcal{D})\right]_{\ell=1}^{L}.$$

This requires only a single forward pass and does not involve dataset-specific gradient updates, posterior sampling, or hyperparameter tuning. Let $g_\mathcal{D}$ denote the monotone time transformation fitted from the observed times in $\mathcal{D}$, and let $\{\mathcal{I}_\ell\}_{\ell=1}^{L}$ denote the discretized bins in the transformed time space. The predicted posterior predictive survival distribution (PPSD) is obtained by summing the probability mass assigned to bins whose raw-time support lies above $t$:

$$\widehat{S}_\omega(t \mid x^*, \mathcal{D}) \;=\; \sum_{\ell: \, g_\mathcal{D}^{-1}(\mathcal{I}_\ell) > t} q_{\omega, \ell}\left(x^*, \widetilde{\delta}^* = 1, \mathcal{D}\right),$$

which is implemented as a cumulative tail sum over the discretized time bins. Similarly, setting $\widetilde{\delta}^* = 0$ yields the PPCD (although we generally do not care about PPCD during inference).

## E. Benchmarking Details

### E.1. Baseline Model Details

In this study, we compare SurvivalPFN against a board range of 20 representative survival analysis methods spanning classical statistical models, tree-based ensembles, discrete-time neural survival models, continuous-time neural survival models, and methods using TFMs. Below, we briefly summarize each baseline, including its core modeling idea and the implementation details used in our benchmark. A side-to-side overview of their methodological properties is provided in Table 2.

For each column in the table, *continuous-time* indicates whether the method explicitly parameterizes a smooth survival distribution over continuous time, without relying on post-hoc interpolation over a discrete time grid. *(Semi-)parametric* indicates whether the method specifies the hazard, density, or survival function through an explicit parametric or semi-parametric form. The *proportional hazards (PH)* assumption indicates that covariates act multiplicatively on a shared baseline hazard, so that hazard ratios between individuals are constant over time. Finally, the *ensemble/mixture* column indicates whether the method combines a finite collection of base learners, components, or experts, such as trees in an ensemble or distributions in a finite mixture model.

In particular, SurvivalMDN is marked parametric because it uses a finite mixture of Gaussians, and DSM is marked parametric because it uses a finite mixture of Weibulls; only an infinite-mixture construction would be nonparametric.

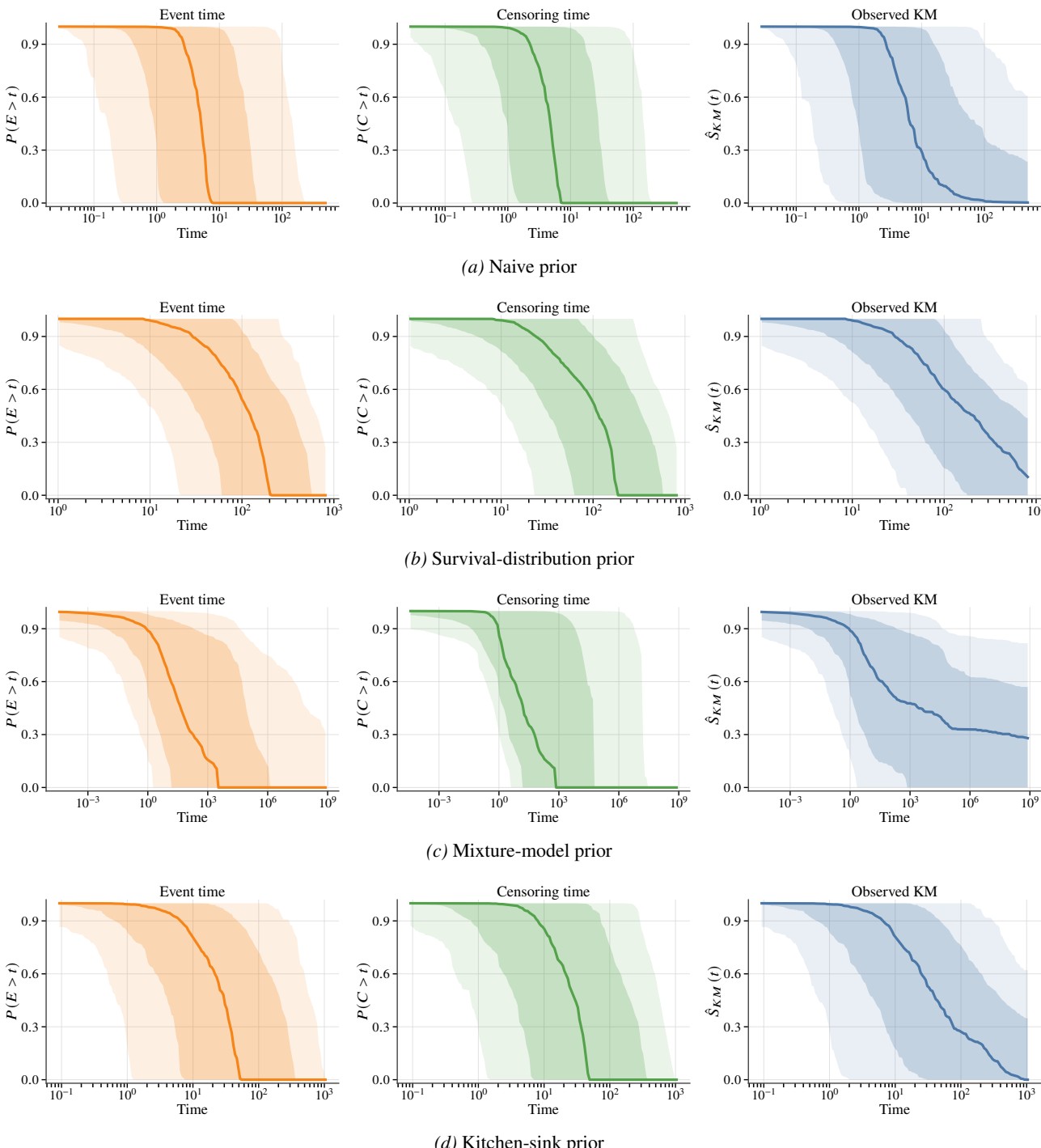

*Figure 8.* **Empirical distributions across 500 synthetic datasets.** Each panel shows the induced marginal event-time survival curve $P(E > t)$, censoring-time survival curve $P(C > t)$, and observed Kaplan-Meier curve $\widehat{S}_{KM}(t)$. Solid lines denote the pointwise median curve across generated datasets. Dark shaded bands denote the interquartile range (25th-75th percentiles), and light shaded bands denote the 10th-90th percentile range.

*Table 2.* Qualitative comparison of survival models considered in our benchmark. Columns indicate whether a method explicitly parameterizes a continuous-time (cont. time) survival distribution, imposes a (semi-)parametric form, assumes proportional hazards (PH), uses an ensemble or finite-mixture construction, or belongs to the tabular foundation model (TMF) family.

| Model | Cont. time | (Semi-) parametric | PH assump. | Ensemble/ mixture | TFM | Distinguishing feature |
|---|---|---|---|---|---|---|
| *Prior-fitted and tabular foundation models* | | | | | | |
| SurvivalPFN | ✗ | ✗ | ✗ | ✗ | ✓ | Novel in-context Bayesian survival estimator. |
| StaticSurvivalTFM | ✗ | ✗ | ✗ | ✗ | ✓ | Handling right-censoring with tabular foundation models. |
| *Classical survival models* | | | | | | |
| CoxPH | ✗ | ✓ | ✓ | ✗ | ✗ | Semi-parametric Cox proportional hazards model. |
| CoxNet | ✗ | ✓ | ✓ | ✗ | ✗ | Regularized Cox proportional hazards model. |
| cSVR | ✗ | ✗ | ✗ | ✗ | ✗ | Censored support-vector regression baseline. |
| *Tree-based models* | | | | | | |
| GB | ✗ | ✓ | ✓ | ✓ | ✗ | Gradient-boosted Cox proportional hazards model. |
| CWGB | ✗ | ✓ | ✓ | ✓ | ✗ | Censoring-weighted gradient-boosted Cox model. |
| RSF | ✗ | ✗ | ✗ | ✓ | ✗ | Random survival forest for nonlinear effects and interactions. |
| *Neural discrete-time models* | | | | | | |
| DeepHit | ✗ | ✗ | ✗ | ✗ | ✗ | Neural discrete-time model emphasizing discrimination. |
| DeepSurv | ✗ | ✓ | ✓ | ✗ | ✗ | Neural Cox proportional hazards model. |
| MTLR | ✗ | ✗ | ✗ | ✗ | ✗ | Multi-task logistic regression for discrete-time survival. |
| Nnet-survival | ✗ | ✗ | ✗ | ✗ | ✗ | Discrete-time neural survival model. |
| CoxTime | ✗ | ✓ | ✗ | ✗ | ✗ | Neural Cox model with time-varying effects. |
| IWSG | ✗ | ✗ | ✗ | ✗ | ✗ | Explicitly models the censoring mechanism. |
| CQRNN | ✗ | ✗ | ✗ | ✗ | ✗ | Quantile-regression survival baseline. |
| BNN-MTLR | ✗ | ✗ | ✗ | ✗ | ✗ | Bayesian neural extension of MTLR. |
| *Neural continuous-time models* | | | | | | |
| DSM | ✓ | ✓ | ✗ | ✓ | ✗ | Finite parametric (Weibull) mixture components. |
| SumoNet | ✓ | ✗ | ✗ | ✗ | ✗ | Continuous survival-function via automatic differentiation. |
| SurvivalMDN | ✓ | ✓ | ✗ | ✓ | ✗ | Finite parametric (Gaussian) mixture components. |
| DeepAFT–Weibull | ✓ | ✓ | ✓ | ✗ | ✗ | Continuous Weibull accelerated failure time model. |
| DeepAFT–Log-logistic | ✓ | ✓ | ✗ | ✗ | ✗ | Continuous log-logistic accelerated failure time model. |

1. **SurvivalPFN**. SurvivalPFN is the model we developed in this paper. See description in Section 3 and Appendix D.

2. **StaticSurvivalTFM** (Kim et al., 2026). StaticSurvivalTFM is a framework that converts any binary classifier into a survival predictor; we provide methodological details in Appendix I. For a fair comparison with SurvivalPFN, we use TabDPT (Ma et al., 2024) as the backbone binary classifier in the main benchmark. We further study the effect of replacing this backbone with MITRA (Zhang et al., 2025) (the reported best version in Kim et al. (2026)), with results reported in Appendix G.4.

3. **Cox proportional hazards model (CoxPH)** (Cox, 1972). CoxPH is the classical semi-parametric proportional-hazards model, estimating covariate effects through the Cox partial likelihood while leaving the baseline hazard unspecified. We used the `scikit-survival` implementation (Pölsterl, 2020), which produces risk scores and can recover survival curves using an estimated baseline hazard (Breslow, 1975).

4. **Elastic-net Cox proportional hazards model (CoxNet)** (Simon et al., 2011). CoxNet regularizes the Cox partial likelihood with an elastic-net penalty, making Cox-style modeling more stable in high-dimensional settings. We used the `scikit-survival` implementation (Pölsterl, 2020), with the same baseline hazard estimator as CoxPH.

5. **Censored Support Vector Regression (cSVR)** (Pölsterl et al., 2015). cSVR extends support-vector regression to right-censored outcomes by treating censored observations through inequality constraints or ranking-style losses. Specifically, we use the `FastSurvivalSVM` method in `scikit-survival` (Pölsterl, 2020). Because cSVR only outputs a scalar regression time rather than a full survival distribution, it cannot produce valid values for evaluation metrics that require an entire survival curve or event-time distribution.

6. **Gradient-boosted Cox model (GB)** (Ridgeway, 1999). GB fits an additive risk model by gradient boosting under a Cox partial-likelihood objective. We used the `scikit-survival` implementation (Pölsterl, 2020), which yields a boosted risk score and survival estimates through the fitted baseline hazard as CoxPH.

7. **Component-wise gradient-boosted Cox model (CWGB)** (Hothorn et al., 2006). CWGB is a component-wise variant of gradient-boosted Cox regression, where weak learners update individual covariate components under a Cox-style objective. We used the `scikit-survival` implementation (Pölsterl, 2020).

8. **Random Survival Forests (RSF)** (Ishwaran et al., 2008). RSF is an ensemble of survival trees that partitions the feature space using survival-specific split criteria and averages terminal-node survival estimates across trees. We use the `scikit-survival` implementation (Pölsterl, 2020).

9. **DeepHit** (Lee et al., 2018). DeepHit is a discrete-time neural survival model that directly parameterizes the probability mass function over event-time bins. We used the `pycox` implementation; its objective combines a likelihood term with a C-index-like ranking term, explicitly encouraging discrimination.

10. **DeepSurv** (Katzman et al., 2018). DeepSurv replaces the linear predictor in CoxPH with a neural network while retaining the Cox partial-likelihood objective and proportional-hazards structure. We reimplemented it following the public PyTorch implementation (`https://github.com/czifan/DeepSurv.pytorch`) and add baseline hazard estimation as in CoxPH.

11. **Multi-task Logistic Regression (MTLR)** (Yu et al., 2011; Fotso, 2018). MTLR parameterizes the survival distribution over a discrete time grid using a sequence of dependent logistic regressors. We reimplemented MTLR following the public PyTorch implementation (`https://github.com/mkazmier/torchmtlr`).

12. **Nnet-survival** (Biganzoli et al., 1998; Gensheimer & Narasimhan, 2019). Nnet-survival models discrete-time conditional hazards with a neural network and trains by a binary cross-entropy-style survival likelihood over time intervals. We used the `pycox` implementation, converting predicted discrete hazards into survival curves for distributional evaluation.

13. **CoxTime** (Kvamme et al., 2019). CoxTime generalizes neural Cox regression by allowing the relative risk to depend on both covariates and time, thereby relaxing the proportional-hazards assumption. We used the `pycox` implementation and recovered survival curves from the learned time-dependent risk function and the same baseline hazard estimator as CoxPH.

14. **Inverse-Weighted Survival Games (IWSG)** (Han et al., 2021). IWSG explicitly models both the failure-time and censoring distributions through an inverse-probability-of-censoring-weighted (IPCW) game objective. We reimplemented it from the official IWSG codebase (https://github.com/rajesh-lab/Inverse-Weighted-Survival-Games).

15. **Censored Quantile Regression Neural Network (CQRNN)** (Pearce et al., 2022). CQRNN directly predicts event-time quantiles under censoring, providing a distribution-free way to represent time-to-event uncertainty. We reimplemented it following the official CQRNN codebase (https://github.com/TeaPearce/Censored_Quantile_Regression_NN) and converted the predicted quantile function into a monotone survival curve when distributional metrics (IBS and D-calibration) were required.

16. **Bayesian Neural Network Multi-task Logistic Regression (BNN-MTLR)** (Qi et al., 2023b). BNN-MTLR extends MTLR with Bayesian neural-network uncertainty, producing a PPD over discrete survival curves. We reimplemented it from the official BNN-MTLR codebase (https://github.com/shi-ang/BNN-ISD).

17. **Deep Survival Machines (DSM)** (Nagpal et al., 2021). DSM parameterizes the event-time distribution as a finite mixture of Weibull components, with neural networks producing mixture weights and component parameters. We reimplemented DSM using the code in `auton-survival` package (Nagpal et al., 2022).

18. **Survival Monotonic Network (SuMoNet)** (Rindt et al., 2022). SuMoNet models continuous-time survival distributions with monotonic neural networks, using automatic differentiation to obtain valid densities from the learned cumulative distribution function. We reimplemented it following the official SuMoNet codebase (https://github.com/MrHuff/Sumo-Net).

19. **Survival Mixture Density Network (SurvivalMDN)** (Han et al., 2022). SurvivalMDN models the event-time distribution using a finite mixture density network, providing a flexible but still finite-dimensional parametric mixture representation. We reimplemented it from the official SurvivalMDN codebase (https://github.com/XintianHan/Survival-MDN).

20. **Neural Weibull accelerated failure-time model (DeepAFT-Weibull)** (Norman et al., 2024). DeepAFT-Weibull uses a neural network to parameterize a Weibull accelerated failure-time distribution for right-censored data. We reimplemented the model following the paper.

21. **Neural log-logistic accelerated failure-time model (DeepAFT-Loglogistic)** (Norman et al., 2024). DeepAFT-Loglogistic similarly uses a neural network to parameterize a log-logistic accelerated failure-time distribution. We reimplemented the model following the paper.

### E.2. Unified Time Grid for Consistent Evaluation

All methods in our benchmark are evaluated through a common prediction object. After fitting, each model is converted into a survival-probability matrix

$$\widehat{\mathbf{S}} = \left[\widehat{S}_i(t_j)\right]_{i=1,\ldots,n_{\text{test}};\, j=1,\ldots,m}, \qquad \mathcal{G} = \{t_1 < \cdots < t_m\},$$

where $\widehat{S}_i(t_j)$ denotes the predicted survival probability for test individual $i$ at time $t_j$. All metrics are then computed from $(\mathcal{G}, \widehat{\mathbf{S}})$. Thus, differences across methods enter only through how their native time representation is constructed before prediction.

To avoid evaluation leakage, all time grids are defined using only the data available during training; test-set event times are never used to define the evaluation support.

For models that require a predefined binning for making discrete-time survival function prediction, including MTLR, DeepHit, Nnet-survival, BNN-MTLR and StaticSurvivalTFM, we use an event-time quantile grid. Let

$$\mathcal{T}_E = \{T_i : \delta_i = 1,\ i \in \mathcal{D}^{tr}\}$$

denote the uncensored event times in the fitting data. When the number of bins is not specified, we set

$$K = \left\lceil \sqrt{|\mathcal{T}_E|} \right\rceil,$$

and define the discrete support by the unique empirical quantiles

$$\mathcal{G}_{\text{disc}} = \text{unique}\left\{ Q_{\mathcal{T}_E}\left(\frac{k-1}{K-1}\right) : k = 1, \dots, K \right\},$$

where $Q_{\mathcal{T}_E}$ is the empirical quantile function. This grid provides the native discretization on which discrete-time models are trained, with each bin contains the exact same number of uncensored instances.

For DeepHit and Nnet-survival, the implementation requries that the first bin location must smaller than the smallest time in the data. Therefore, we additionally shift the first bin location:

$$b_1 \leftarrow \max\left\{ \min_{i \in \mathcal{D}^{tr}}\{T_i - \epsilon\}, 0 \right\}, \qquad \epsilon = 10^{-5}.$$

Continuous-time models do not require a training-time discretization. Nevertheless, curve-based evaluation still requires a finite query set. For these methods, we define the evaluation support from the observed fitting durations:

$$\mathcal{G}_{\text{cont}} = \{0\} \cup \text{unique}\{T_i : i \in \mathcal{D}^{tr}\}, \tag{E.1}$$

with duplicate zeros removed if necessary. For quantile-output models such as CQRNN, predicted quantile functions are first converted to survival probabilities on the same support before metric computation.

This protocol preserves each model class's natural time parameterization: discrete-time methods learn on event-time quantile bins, whereas continuous-time methods are queried on the empirical training-time support.

### E.3. Hyperparameter Tuning

We tune hyperparameters only for neural-network baselines whose performance depends on optimizer and architecture choices. Classical `scikit-survival` models (CoxPH, CoxNet, cSVR, GB, CWGB, and RSF), StaticSurvivalTFM, and SurvivalPFN are kept fixed at their benchmark settings.

For each outer split $r$, hyperparameter selection is performed using only the training-side data,

$$\mathcal{D}_{(r)}^{tr+val} = \mathcal{D}_{(r)}^{tr} \cup \mathcal{D}_{(r)}^{val},$$

and the test set is never used for either tuning or discretization. For each model $m$, we sample $R = 10$ configurations from its search space $\Lambda_m$ and estimate their performance by shuffled $F = 5$-fold cross-validation on $\mathcal{D}_{(r)}^{tr+val}$, using the outer split seed for reproducibility. The selected configuration is

$$\lambda_{m,(r)}^* = \arg\min_{\lambda \in \{\lambda_1, \dots, \lambda_R\}} \frac{1}{F} \sum_{f=1}^{F} \mathcal{L}_{(f)}^{val}(\lambda; m),$$

where $\mathcal{L}_{(f)}^{val}$ denotes the model-specific objective loss on the validation on fold $f$. Configurations that fail to complete training are treated as infeasible and assigned infinite validation loss. After selection, model $m$ is refit on $\mathcal{D}_{(r)}^{tr+val}$ using $\lambda_{m,(r)}^*$, and evaluated once on the held-out test split.

The full model-specific search spaces are summarized in Table 3. All remaining optimization settings are fixed across tuning trials: AdamW optimizer, batch size 256, ReLU activations, no normalization layer, early stopping, and a maximum budget of 10,000 epochs. For models that require an explicit learning-rate floor, we set

$$\eta_{\min} = 10^{-3}\eta,$$

where $\eta$ is the tuned learning rate. Model-specific constants not included in Table 3 are fixed throughout tuning.

### E.4. Evaluation Metrics

We evaluate the model's performance on the held-out test set $\mathcal{D}^{te}$. For a fitted model, let $\widehat{S}_{E|X}(u \mid x_i)$ denote the predicted event-time survival function for subject $i$. When a point prediction is required, we use the predicted median survival time

$$\widehat{e}_i = \inf\left\{ u : \widehat{S}_{E|X}(u \mid x_i) \leq 1/2 \right\},$$

*Table 3.* Hyperparameter search spaces for tuned neural baselines. The table defines the shared base space and model-specific extensions.

| Models | Hyperparameter meanings | Search space |
|---|---|---|
| *Base hyperparameters* | | |
| DeepHit DeepSurv MTLR Nnet-survival CoxTime DeepAFT-Weibull DeepAFT-Loglogistic | `lr`: learning rate; `weight_decay`: weight decay; `neurons`: hidden architecture; `dropout`: dropout probability | { `lr`: $\{10^{-4}, 10^{-3}, 10^{-2}\}$; `weight_decay`: $\{10^{-3}, 10^{-2}, 10^{-1}\}$; `neurons`: $\{[64], [64, 32], [64, 64, 16], [32], [32, 16], [32, 32, 16], [16], [16, 8], [8], []\}$; `dropout`: $\{0.0, 0.4, 0.6\}$ } |
| *Model-specific hyperparameters* | | |
| CQRNN | `n_quantiles`: number of predicted quantile levels. | { Base; `n_quantiles`: $\{9, 19, 39\}$ } |
| SumoNet | `neurons_alter`: hidden architecture for the censoring branch. | { Base; `neurons_alter` = `neurons` } |
| IWSG | `neurons_alter`: hidden architecture for the censoring branch. | { Base; `neurons_alter` = `neurons` } |
| SurvivalMDN | `n_mixtures`: number of mixture components. | { Base; `n_mixtures`: $\{3, 5, 10, 50\}$ } |
| DSM | `n_mixtures`: number of mixture components. | { Base; `n_mixtures`: $\{3, 5, 10, 50\}$ } |
| BNN-MTLR | `pi`: prior mixture probability. | { Base; `pi`: $\{0.2, 0.5, 0.8\}$ } |

and define the corresponding risk score as

$$\widehat{r}_i = -\widehat{e}_i,$$

so that higher risk corresponds to shorter predicted survival.

**Concordance Index.** Discrimination measures whether a model correctly orders subjects by risk. We use Harrell's concordance index (Harrell et al., 1982), computed over comparable pairs in which subject $i$ experiences an observed event before subject $j$:

$$\text{CI} = \frac{\sum_{i=1}^{n} \sum_{j \neq i} \delta_i \, \mathbb{1}[t_i < t_j] \, \mathbb{1}[\widehat{r}_i > \widehat{r}_j]}{\sum_{i=1}^{n} \sum_{j \neq i} \delta_i \, \mathbb{1}[t_i < t_j]}.$$

Higher CI values indicate better performance.

**Integrated Brier Score.** The Brier score evaluates probabilistic accuracy at a target time $u$. Because the event status at $u$ may be unknown for subjects censored before $u$, we use inverse-probability-of-censoring weighting (IPCW) (Graf et al., 1999; Robins & Finkelstein, 2000). Let $\widehat{S}_C$ denote the Kaplan-Meier estimate of the marginal censoring survival function, estimated from the training-side data. The IPCW Brier score is

$$\text{BS}(u) = \frac{1}{n} \sum_{i=1}^{n} \left[ \frac{\delta_i \mathbb{1}[t_i \leq u] \cdot \widehat{S}_{E|X}(u \mid x_i)^2}{\widehat{S}_C(t_i)} + \frac{\mathbb{1}[t_i > u] \left(1 - \widehat{S}_{E|X}(u \mid x_i)\right)^2}{\widehat{S}_C(u)} \right].$$

The integrated Brier score averages this quantity over an evaluation horizon $\tau$:

$$\text{IBS} = \frac{1}{\tau} \int_0^{\tau} \text{BS}(u) \, du. \tag{E.2}$$

In our experiments, $\tau$ is chosen from the training-side event-time support, so that the evaluation horizon is not determined by the held-out test set. Lower IBS indicates better performance.

**Mean Absolute Error.** To evaluate point prediction of event time, we use the mean absolute error based on pseudo-observations (MAE-PO) (Qi et al., 2023a). For uncensored subjects, the observed event time $t_i$ can be used directly. For censored subjects, it constructs a pseudo-observation $\widetilde{e}_i$ that estimates the subject's contribution to the marginal Kaplan-Meier survival curve, and then weights the corresponding error by a confidence weight $w_i$. The resulting error takes the form

$$\text{MAE-PO} \;=\; \frac{\sum_{i=1}^{n} w_i \, |\widehat{e}_i - \widetilde{e}_i|}{\sum_{i=1}^{n} w_i}.$$

This produces an event-time error metric that can use both uncensored and censored test subjects. Lower MAE value indicates better performance.

**Distribution Calibration.** Distribution calibration (D-calibration) evaluates whether the predicted survival distribution is calibrated over the full time axis (Haider et al., 2020). For an uncensored subject, define the probability integral transform value

$$u_i \;=\; \widehat{S}_{E|X}(t_i \mid x_i).$$

If the predicted survival distributions are calibrated, then $\{u_i : \delta_i = 1\}$ should follow a standard uniform distribution on $[0, 1]$. In practice, we partition the probability range $[0, 1]$ into 10 equal-width bins. Uncensored subjects contribute to the bin containing $\widehat{S}_{E|X}(t_i \mid x_i)$. For censored subjects, the event time is only known to satisfy $e_i > t_i$, so their contribution is "blurred" over the portion of the probability scale consistent with this information, namely $\left[0, \widehat{S}_{E|X}(t_i \mid x_i)\right]$. D-calibration is then assessed by the chi-square statistic over the numbers in all the bins. We report the chi-square statistic for each experiments, a smaller statistic indicates less evidence against distribution calibration.

**Log-rank Reliability Test.** We also use a log-rank goodness-of-fit test to assess whether predicted event times are statistically aligned with the observed time-to-event data. The test compares the observed test sample

$$\mathcal{A}_{\text{obs}} \;=\; \{(t_i, \delta_i)\}_{i=1}^{n}$$

with the predicted event-time sample

$$\mathcal{A}_{\text{pred}} \;=\; \left\{\left(\widehat{e}_i, \widehat{\delta}_i = 1\right)\right\}_{i=1}^{n},$$

where predicted median survival times are treated as uncensored event-time predictions. We report the log-rank statistic for each experiments, a smaller statistic indicate closer agreement between the predicted event-time distribution and the observed test outcomes.

All metrics are computed using the `SurvivalEVAL` package (Qi et al., 2023c).

### E.5. Compute Environment and Runtime Protocol

All benchmark experiments were executed on a Slurm-managed GPU cluster. Each experiment was submitted as an independent job with a fixed resource budget: one NVIDIA L40S GPU (48 GB memory), one CPU core from Intel Xeon Gold 6448Y, 16 GB of system memory, and a maximum wall-clock time of 72 hours. The same resource allocation was used across models and datasets for fair evaluation.

The 72-hour wall-clock limit includes all computation required for a benchmark run, including model fitting, hyperparameter tuning when enabled, final refitting, prediction on the held-out test set, and metric computation. If a run did not complete successfully within this budget, we treated the corresponding model-dataset run as failure. Failed runs were deemed as the worst (or equally worst if multiple models failed on this dataset) in the ranking procedure described in Appendix E.6.

### E.6. Performance Reporting Protocol

We evaluate each model on repeated random train/validation/test splits. Each dataset is evaluated over $R = 10$ repetitions. The experiment seed is set from 0 to 9 to these repetition, to support data split and model initialization (if needed).

For each split $r$, the dataset is divided into a 70% $\mathcal{D}^{tr+val}_{(r)}$ and a 30% $\mathcal{D}^{te}_{(r)}$. We compute performance scores on the held-out test set via the metrics described in Appendix E.4. We report the mean and standard deviation across 10 split.

For dataset-level comparisons, each model is ranked separately within each dataset and metric. Rank 1 is assigned to the best value, with ties receiving the minimum tied rank. If a model does not produce a valid result for a given dataset-metric pair, it is assigned one rank below the worst completed method for that pair.

The cSVR baseline produces only a scalar event-time prediction rather than a full survival distribution. Consequently, distributional metrics such as IBS and D-calibration are not applicable to cSVR on any dataset.

Across the full benchmark, we evaluated 22 models on 61 datasets, yielding $21 \times 61 = 1281$ model-dataset runs. Among these, 28 runs failed to produce valid results, corresponding to a failure rate of 2.19%. We summarize the failures below.

- **CoxPH** failed on 13 datasets: micro.censure, nki70, stagec, BMT, cancer, zinc, BCCardiotox, vlbw, PDM, actg, METABRIC, AIDS, and hdfail. In all cases, the failure was caused by a singular Hessian matrix of the Cox partial log-likelihood, which prevented the Newton-Raphson optimizer from computing a valid update.[2]

- **DeepSurv** failed on 1 dataset, PDM. The fitted baseline hazard produced a degenerate survival curve equal to 1 at all time points, resulting in identical predictions for all test instances.

- **CoxTime** failed on 1 dataset, FRTCS, because the predicted survival matrix contained NaN values.

- **IWSG** failed on 5 datasets: SUPPORT, MIMIC-IV_all, hdfail, SEER_brain, and SEER_liver. In all cases, the model did not complete training within the 72-hour wall-clock limit.

- **DSM** failed on 3 datasets: FRTCS, NPC, and MIMIC-IV_all. For FRTCS and NPC, the predicted survival matrix contained NaN values. For MIMIC-IV_all, the run did not complete within the 72-hour wall-clock limit.

- **SurvivalMDN** failed on 3 datasets: SEER_liver, SEER_brain, and hdfail. On hdfail, the run exceeded the available 48 GB GPU memory. On SEER_liver and SEER_brain, the model did not complete within the 72-hour wall-clock limit.

- **DeepAFT-Weibull** failed on 1 dataset, FRTCS, because the model did not converge during training.

- **DeepAFT-Loglogistic** failed on 1 dataset, FRTCS, because the model did not converge during training.

Let $\mathrm{rank}_{m,\mathcal{D},q}$ denote the rank of model $m$ on dataset $\mathcal{D}$ for metric $q$. For each model and metric, we summarize performance by the median rank across datasets,

$$\widetilde{r}_{m,q} \;=\; \mathrm{median}_{\mathcal{D}}\, \mathrm{rank}_{m,\mathcal{D},q}\,.$$

To quantify uncertainty in this median rank, we use a nonparametric bootstrap over datasets. Specifically, for each bootstrap replicate $b = 1, \ldots, B$, we sample datasets with replacement from the benchmark collection, recompute the median rank,

$$\widetilde{r}^{(b)}_{m,q} \;=\; \mathrm{median}_{\mathcal{D} \in \mathcal{B}^{(b)}}\, \mathrm{rank}_{m,\mathcal{D},q},$$

and report the 95% bootstrap confidence interval as

$$\left[ Q_{0.025}\left(\{\widetilde{r}^{(b)}_{m,q}\}^B_{b=1}\right), Q_{0.975}\left(\{\widetilde{r}^{(b)}_{m,q}\}^B_{b=1}\right)\right].$$

The overall rank is computed by pooling ranks over all evaluation metrics before applying the same median-rank and bootstrap procedure.

## F. Benchmark Dataset Details

We construct a large collection of real-world survival datasets from two sources. First, we use datasets from the `SurvSet` package (Drysdale, 2022). We exclude datasets that are longitudinal, contain competing risks rather than a single event of interest, or are high-dimensional in the sense that the number of features exceeds the number of samples. After applying

---

[2]This issue can often be mitigated by adding regularization. However, because we include CoxNet as the regularized Cox baseline, we intentionally keep CoxPH as the conventional unregularized Cox model.

these criteria, 57 `SurvSet` datasets remain. Second, we collect 24 additional survival datasets from textbooks, recent papers, and public software packages, restricting attention to datasets with at most 100,000 samples. This yields a total of 81 real-world datasets, which, to our knowledge, constitutes one of the largest survival-analysis benchmark collections considered in a single study. Summary statistics for all datasets are provided in Table 4.

Among the 81 datasets, we designate 20 datasets as validation datasets for selecting the SurvivalPFN checkpoint during training.[3] The remaining 61 datasets are held out for the final benchmark and are not used for checkpoint selection. For each validation dataset, we use a deterministic split with 70% of samples as the in-context training set and 30% as the inference set. At each training epoch, the current SurvivalPFN checkpoint is evaluated on all 20 validation datasets. We select the checkpoint with the best weighted average integrated Brier score,

$$\text{Weighted-IBS}(\theta) = \frac{\sum_{\mathcal{D}} \sqrt{N} \, \text{IBS}_{\mathcal{D}}(\theta)}{\sum_{\mathcal{D}} \sqrt{N}}$$

$$\theta^* = \arg\min_{\theta} \text{Weighted-IBS}(\theta),$$

where $\text{IBS}_{\mathcal{D}}(\theta)$ is the IBS of checkpoint $\theta$ on the inference split of dataset $\mathcal{D}$, calculated using Equation E.2. The square-root weighting gives larger datasets more influence while avoiding domination by the largest cohorts.

The validation datasets were chosen to cover a broad range of empirical regimes. In particular, we considered sample size, censoring rate, and the tail survival probability estimated by the Kaplan-Meier curve, $\widehat{S}_{\text{KM}}(t_{\max})$.

*Table 4.* Summary of survival datasets used for checkpoint selection and held-out benchmarking. Features (cat. feat.) reports the total number of covariates, with the number of categorical covariates in parentheses.

| Dataset | Number of samples | Features (cat. feat.) | Missing features | Missing rate | Censoring rate | $\widehat{S}(t_{\max})$ |
|---|---|---|---|---|---|---|
| *Validation datasets for checkpoint selection* | | | | | | |
| ovarian | 26 | 4 (4) | 0 | 0.0% | 53.8% | 49.7% |
| glioma | 37 | 4 (4) | 0 | 0.0% | 37.8% | 34.9% |
| leukemia | 42 | 3 (2) | 0 | 0.0% | 28.6% | 18.9% |
| pharmacoSmoking | 125 | 16 (16) | 0 | 0.0% | 28.8% | 28.8% |
| d.oropha.rec | 192 | 24 (24) | 0 | 0.0% | 27.6% | 16.5% |
| Pbc3 | 349 | 19 (19) | 4 | 0.4% | 82.5% | 63.4% |
| retinopathy | 394 | 11 (11) | 0 | 0.0% | 60.7% | 53.1% |
| Rossi | 432 | 7 (5) | 0 | 0.0% | 73.6% | 73.6% |
| phpl04K8a | 442 | 21 (21) | 0 | 0.0% | 46.6% | 0.0% |
| prostate | 502 | 25 (25) | 4 | 0.2% | 29.5% | 23.8% |
| uis | 628 | 14 (14) | 3 | 0.6% | 19.1% | 16.6% |
| grace | 1,000 | 5 (5) | 0 | 0.0% | 67.6% | 63.5% |
| rdata | 1,040 | 6 (6) | 0 | 0.0% | 47.4% | 38.1% |
| TRACE | 1,878 | 6 (6) | 0 | 0.0% | 49.0% | 43.9% |
| Aids2 | 2,839 | 12 (12) | 0 | 0.0% | 38.0% | 5.8% |
| UnempDur | 3,241 | 6 (6) | 0 | 0.0% | 38.7% | 10.5% |
| smarto | 3,873 | 34 (34) | 16 | 4.8% | 88.1% | 72.5% |
| dataDIVAT1 | 5,943 | 16 (16) | 0 | 0.0% | 83.6% | 0.0% |
| oldmort | 6,495 | 14 (14) | 0 | 0.0% | 69.7% | 0.0% |
| prostateSurvival | 14,294 | 6 (6) | 0 | 0.0% | 94.4% | 81.4% |
| *Held-out test datasets for benchmarking* | | | | | | |
| Bergamaschi | 82 | 10 (10) | 0 | 0.0% | 65.9% | 58.3% |
| larynx | 90 | 4 (3) | 0 | 0.0% | 44.4% | 29.7% |

*Continued on next page*

---

[3]SurvivalPFN is not trained or fine-tuned on these validation datasets, nor on any other real-world survival dataset; they are used only for checkpoint selection.

| Dataset | Number of samples | Features (cat.) | Missing feat. | Missing rate | Censoring rate | $\widehat{S}(t_{\max})$ |
|---|---|---|---|---|---|---|
| lupus | 95 | 5 (2) | 1 | 0.2% | 70.5% | 36.8% |
| micro.censure | 117 | 81 (81) | 0 | 0.0% | 77.8% | 42.4% |
| cgd | 128 | 23 (23) | 0 | 0.0% | 65.6% | 47.6% |
| veteran | 137 | 8 (8) | 0 | 0.0% | 6.6% | 0.0% |
| nki70 | 144 | 76 (76) | 0 | 0.0% | 66.7% | 48.0% |
| stagec | 146 | 18 (18) | 2 | 0.4% | 63.0% | 55.8% |
| burn | 154 | 13 (13) | 0 | 0.0% | 68.8% | 48.0% |
| BMT | 187 | 39 (24) | 11 | 1.1% | 54.5% | 53.3% |
| WPBC | 198 | 32 (1) | 1 | 0.1% | 76.3% | 65.3% |
| Melanoma | 205 | 5 (5) | 0 | 0.0% | 72.2% | 64.5% |
| hepatoCellular | 227 | 43 (43) | 26 | 33.6% | 57.3% | 51.2% |
| cancer | 228 | 28 (28) | 4 | 1.0% | 27.6% | 5.0% |
| NCCTG | 228 | 8 (1) | 6 | 3.7% | 27.6% | 5.0% |
| mgus | 241 | 12 (12) | 4 | 8.7% | 6.6% | 1.8% |
| e1684 | 284 | 3 (3) | 0 | 0.0% | 31.0% | 28.3% |
| HFCR | 299 | 11 (5) | 0 | 0.0% | 67.9% | 57.6% |
| ova | 358 | 11 (11) | 0 | 0.0% | 25.7% | 22.2% |
| diabetes | 394 | 4 (4) | 0 | 0.0% | 60.7% | 53.0% |
| PBC | 418 | 17 (7) | 12 | 14.5% | 61.5% | 35.3% |
| zinc | 431 | 20 (20) | 0 | 0.0% | 81.2% | 79.0% |
| Unemployment | 452 | 7 (7) | 0 | 0.0% | 43.4% | 5.8% |
| whas500 | 461 | 17 (17) | 0 | 0.0% | 61.8% | 0.0% |
| cost | 518 | 13 (13) | 0 | 0.0% | 22.0% | 22.0% |
| BCCardiotox | 531 | 25 (15) | 22 | 5.7% | 89.8% | 69.0% |
| GBM | 591 | 10 (7) | 3 | 4.5% | 17.1% | 0.0% |
| vlbw | 617 | 41 (41) | 5 | 1.7% | 82.7% | 0.0% |
| GBSG2 | 686 | 8 (2) | 0 | 0.0% | 56.4% | 34.3% |
| FRTCS | 697 | 13 (13) | 2 | 0.1% | 89.7% | 57.2% |
| kidney_transplant | 863 | 4 (3) | 0 | 0.0% | 83.8% | 72.4% |
| dataOvarian1 | 912 | 162 (162) | 0 | 0.0% | 40.4% | 11.9% |
| colon | 929 | 12 (12) | 1 | 0.2% | 51.3% | 45.5% |
| credit | 1,000 | 31 (20) | 1 | 0.6% | 30.0% | 5.9% |
| PDM | 1,000 | 8 (5) | 0 | 0.0% | 60.3% | 0.0% |
| LeukSurv | 1,043 | 29 (29) | 0 | 0.0% | 15.7% | 4.4% |
| actg | 1,151 | 17 (17) | 0 | 0.0% | 91.7% | 90.1% |
| COVID | 1,422 | 9 (2) | 0 | 0.0% | 90.5% | 1.1% |
| WHAS | 1,638 | 6 (4) | 0 | 0.0% | 57.9% | 35.7% |
| dataDIVAT2 | 1,837 | 4 (4) | 0 | 0.0% | 68.3% | 31.1% |
| scania | 1,931 | 8 (8) | 0 | 0.0% | 43.8% | 15.9% |
| churn | 1,958 | 19 (10) | 0 | 0.0% | 52.4% | 24.4% |
| METABRIC | 1,981 | 79 (73) | 0 | 0.0% | 55.2% | 11.6% |
| AIDS | 2,139 | 22 (14) | 0 | 0.0% | 24.4% | 0.0% |
| NACD | 2,396 | 48 (33) | 0 | 0.0% | 36.4% | 12.5% |
| rott2 | 2,982 | 12 (12) | 0 | 0.0% | 57.3% | 26.5% |
| divorce | 3,371 | 4 (4) | 0 | 0.0% | 69.4% | 55.7% |
| acath | 3,504 | 3 (3) | 1 | 11.9% | 33.4% | 0.0% |
| NWTCO | 4,028 | 6 (5) | 0 | 0.0% | 85.8% | 84.9% |
| dataDIVAT3 | 4,267 | 16 (16) | 0 | 0.0% | 94.4% | 84.7% |
| Framingham | 4,699 | 17 (17) | 2 | 0.1% | 68.7% | 60.5% |
| NPC | 6,449 | 9 (3) | 0 | 0.0% | 80.8% | 75.6% |

*Continued on next page*

| Dataset | Number of samples | Features (cat.) | Missing feat. | Missing rate | Censoring rate | $\widehat{S}(t_{\max})$ |
|---|---|---|---|---|---|---|
| Dialysis | 6,805 | 72 (72) | 0 | 0.0% | 76.4% | 58.7% |
| FLCHAIN | 7,871 | 23 (2) | 1 | 0.7% | 72.5% | 68.2% |
| MSKCC | 8,130 | 206 (199) | 2 | 0.0% | 70.3% | 34.8% |
| SUPPORT | 9,105 | 31 (11) | 10 | 4.0% | 31.9% | 24.1% |
| employee | 11,991 | 16 (9) | 0 | 0.0% | 83.4% | 50.8% |
| MIMIC-IV_all | 38,520 | 91 (6) | 0 | 0.0% | 66.7% | 0.0% |
| hdfail | 52,422 | 88 (88) | 0 | 0.0% | 94.5% | 56.1% |
| SEER_brain | 73,703 | 10 (4) | 0 | 0.0% | 40.1% | 26.6% |
| SEER_liver | 82,841 | 14 (1) | 0 | 0.0% | 37.6% | 18.0% |

We briefly describe below the 24 datasets that are not taken from `SurvSet`. For the `SurvSet` datasets, please refer to Drysdale (2022).

- *leukemia*: The *leukemia* dataset records remission survival for 42 patients with sex, treatment assignment, and log white blood cell count (Kleinbaum & Klein, 1996).

- *larynx*: The *larynx* cancer dataset follows 90 male patients from first treatment until death or study end, with disease stage, age, and diagnosis year (Kardaun, 1983).

- *lupus*: The *lupus* dataset records survival times and diagnostic features for systemic lupus erythematosus patients (Merrell & Shulman, 1955).

- *BMT*: The Bone Marrow Transplant (*BMT*) Children dataset describes pediatric patients with malignant and nonmalignant hematologic diseases who underwent unrelated-donor allogeneic hematopoietic stem cell transplantation (Sikora et al., 2019).

- *NCCTG*: The *NCCTG* lung cancer dataset follows advanced lung cancer patients with physician-rated and patient-rated performance scores (Loprinzi et al., 1994).

- *HFCR*: The Heart Failure Clinical Records (*HFCR*) dataset contains follow-up outcomes and clinical measurements for 299 heart failure patients (Chicco & Jurman, 2020).

- *Rossi*: The *Rossi* dataset follows 432 Maryland prison releasees for one year to study recidivism after financial-aid treatment assignment (Rossi et al., 1980).

- *BCCardiotox*: *BCCardiotox* follows HER2+ breast cancer patients treated with potentially cardiotoxic therapies and records time to cancer therapy-related cardiac dysfunction (Pineiro-Lamas et al., 2023).

- *GBM*: The TCGA glioblastoma multiforme (*GBM*) dataset contains clinical survival information for primary brain tumor patients (source sites: Duke University Medical School McLendon Roger 1 Friedman Allan 2 Bigner Darrell 1 et al., 2008).

- *kidney_transplant*: The *kidney_transplant* dataset records post-transplant death times with recipient age, gender, and race (Klein & Moeschberger, 2003).

- *credit*: The `PySurvival` (Fotso et al., 2019–) credit-risk dataset adapts the German Credit data to model the time until a loan is fully repaid.

- *PDM*: The `PySurvival` (Fotso et al., 2019–) predictive-maintenance (*PDM*) dataset models the time until an industrial machine breaks.

- *COVID*: The COVID-19 Asian discharge dataset models time to hospital discharge for patients with COVID-19 diagnosis (Kumar et al., 2022).

- *WHAS*: The Worcester Heart Attack Study (*WHAS*) follows acute myocardial infarction patients after hospital admission (Hosmer Jr et al., 2008).

- *churn*: The `PySurvival` (Fotso et al., 2019–) churn dataset models when SaaS customers stop their monthly subscription.

- *METABRIC*: *METABRIC* profiles primary breast tumors with long-term clinical follow-up for breast-cancer prognosis (Curtis et al., 2012).

- *AIDS*: ACTG 175 records HIV-infected adults randomized to nucleoside monotherapy or combination therapy (Hammer et al., 1996).

- *NACD*: The Northern Alberta Cancer Dataset contains clinical records for several cancer sites, which used to predict the death from cancer onset (Yu et al., 2011).

- *NPC*: The nasopharyngeal carcinoma dataset follows patients from Sun Yat-sen University Cancer Center for progression-free survival after radiotherapy with or without chemotherapy (Tang et al., 2016). Original training and validation cohorts are combined.

- *MSKCC*: The MSK-IMPACT cohort contains targeted sequencing and clinical annotations from more than 10,000 advanced cancer cases (Zehir et al., 2017).

- *MIMIC-IV_all*: *MIMIC-IV_all* is derived from the MIMIC-IV critical care database, which contains de-identified electronic health records for patients admitted to intensive care units (Johnson et al., 2023). We use the all-cause mortality cohort curated by Qi et al. (2023a), restricting to patients who survived at least 24 hours after ICU admission.

- *employee*: The `PySurvival` (Fotso et al., 2019–) employee-retention dataset models when employees leave a company using HR and workload attributes.

- *SEER_brain*: *SEER_brain* is a brain cancer cohort derived from the Surveillance, Epidemiology, and End Results (SEER) Program registry. The task is to predict time from cancer diagnosis to death or censoring, using the cohort curated by Qi et al. (2024).

- *SEER_liver*: *SEER_liver* is the corresponding liver cancer cohort from SEER, with the same time-to-death prediction target and curation protocol (Qi et al., 2024).

## G. Additional Experimental Results

### G.1. RQ1: Predictive Performance

Figure 5 has demonstrate the overall performance across 61 benchmark datasets. The intervals reflect uncertainty in the estimated median rank across the benchmark collection, rather than the variability of ranks themselves. In this appendix, we further analyze the benchmark results by stratifying datasets according to sample size and censoring rate.

The sample-size stratification reveals that SurvivalPFN is particularly effective in the small-data regime. On small datasets (Figure 9), SurvivalPFN is clearly separated from most baselines in overall rank and performs strongly across all five metrics, including IBS, CI, D-calibration, MAE, and Log-Rank. Traditional statistical survival methods and tree-based methods performance in the second tier, while neural-network-based methods generally performance the worst.

For medium-sized datasets (Figure 10), SurvivalPFN remains among the strongest methods overall, with especially competitive performance on IBS, Log-Rank. However, the gap to the best baselines becomes smaller, and several nerual-network-based methods (*e.g.*, MTLR, SurvivalMDN) become competitive on the overall performance and also on individual metrics such as CI or D-calibration.

On large datasets (Figure 11), the advantage of SurvivalPFN decreases further: its overall, IBS, and CI ranks move leftward compared with the small-data setting, although it remains competitive on MAE and Log-Rank. This trend suggests a size-regime trade-off. As more observations become available, deep-learning-based estimators can benefit more directly from the larger sample size, whereas SurvivalPFN is constrained by finite context length and the need to summarize large tables during inference.

In contrast, the censoring-rate stratification shows substantially more stable behavior. Across low-, medium-, and high-censoring subsets (Figures 12-14), SurvivalPFN remains in the top group overall and retains strong performance on IBS, MAE, and Log-Rank. This consistency is encouraging because censoring affects the available event-time information

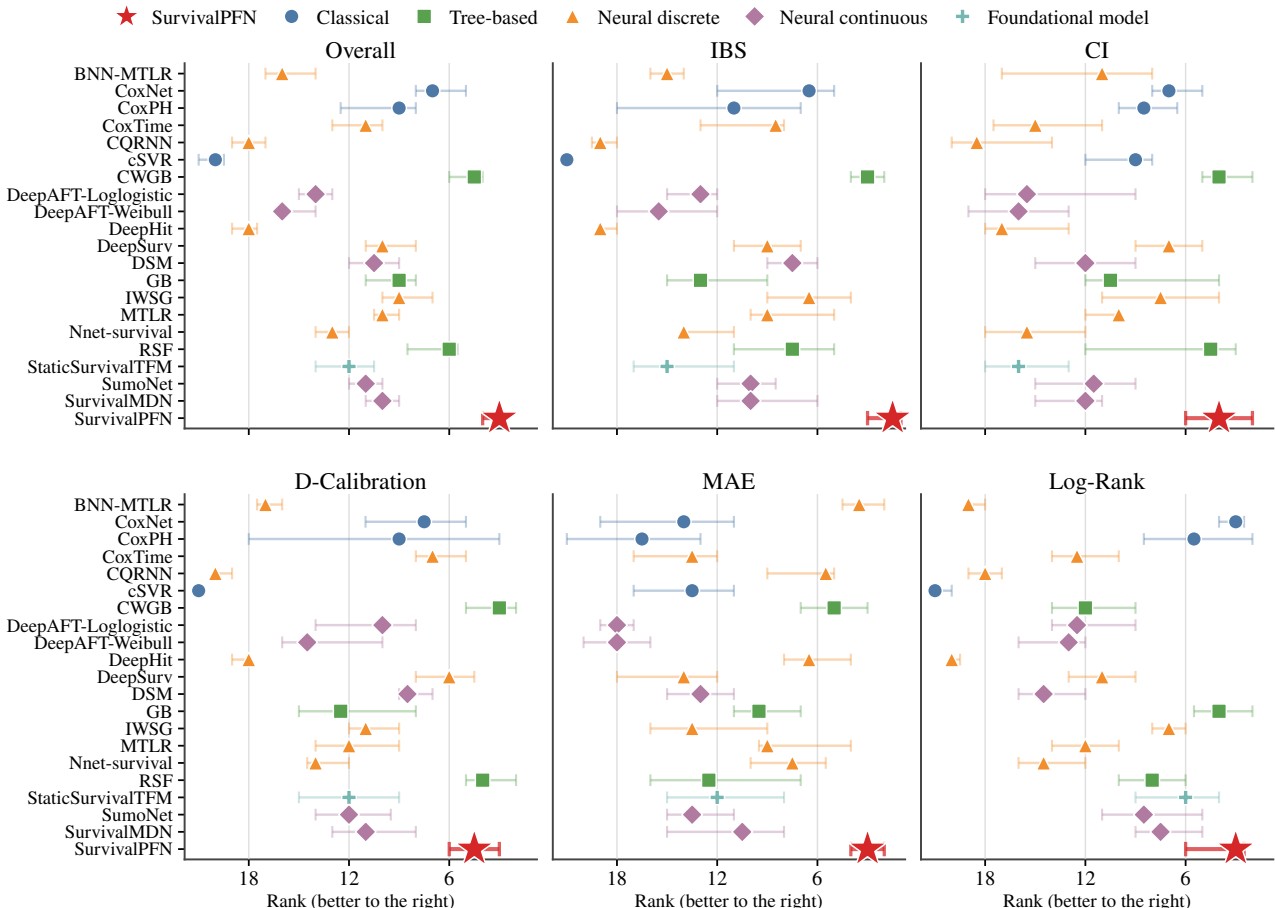

*Figure 9.* **Model ranks across 24 small-size datasets** ($N < 500$). Points/stars denote median ranks across datasets, with horizontal bars showing 95% bootstrap confidence intervals for the median rank.

and can differentially impact discrimination, calibration, and distributional accuracy metrics. The results suggest that the synthetic prior used to train SurvivalPFN provides useful robustness across censoring regimes. Although the strongest baseline varies across metrics and censoring levels, no single baseline matches SurvivalPFN's overall consistency across dataset strata, censoring strata, and evaluation metrics.

### G.2. RQ2: Computational Efficiency

For the runtime comparison in Figure 1, datasets with at least one failed run are excluded from this runtime aggregation, so every method is compared on the same set of completed datasets. This prevents methods from appearing artificially faster because they crashed, timed out, or failed to produce valid predictions on more demanding datasets. Predictive-performance rankings still follow the failure handling protocol in Appendix E.6.

### G.3. RQ3: Sensitivity to Training-Set Size

Figure 15 extends the training-ratio analysis to 16 additional datasets. Across these datasets, increasing the training ratio improves all methods, with IBS decreasing and CI increasing most sharply when moving from very small context sizes to moderate context sizes. SurvivalPFN remains stable across ratios and is frequently among the best methods on both metrics, especially in low-data regimes.

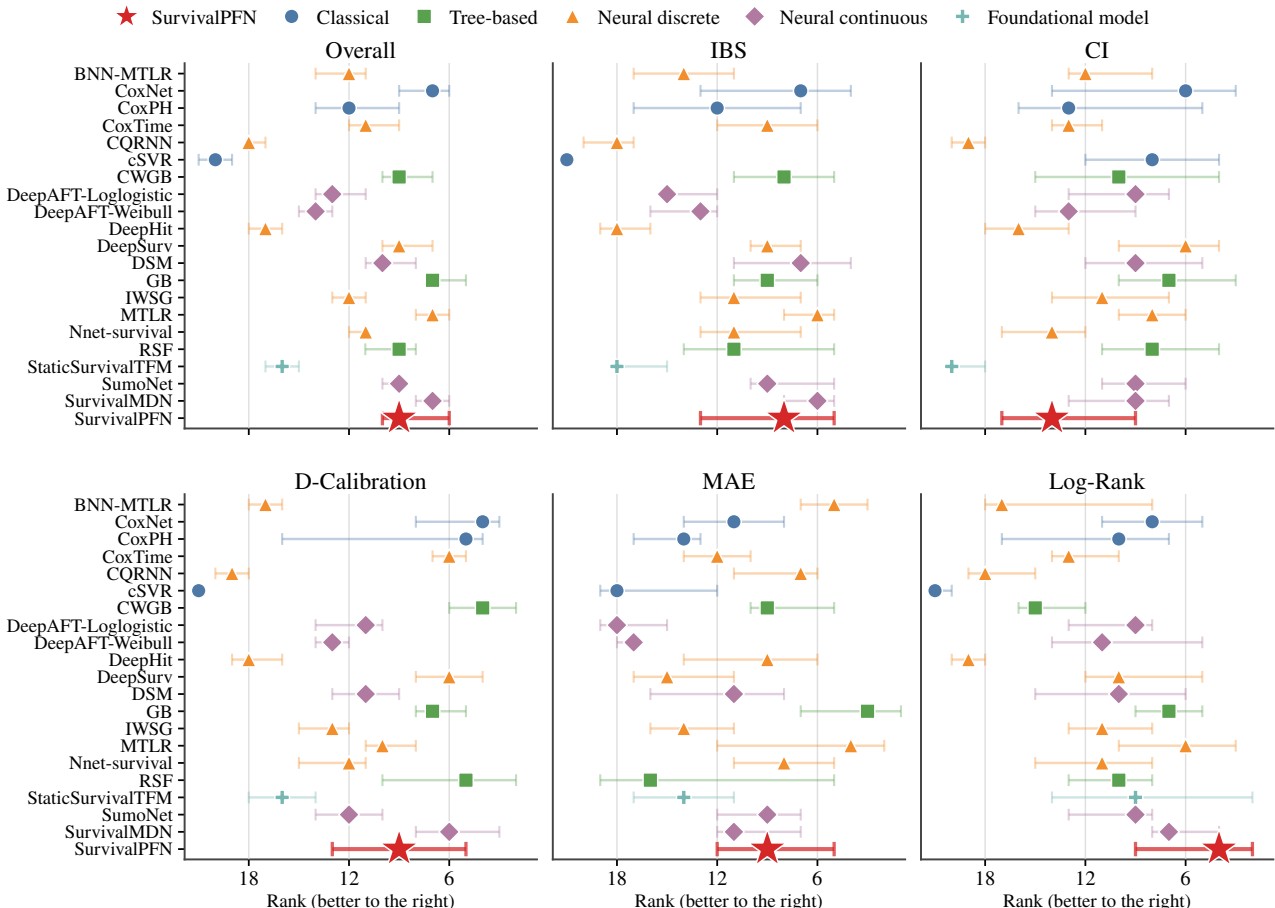

*Figure 10.* **Model ranks across 27 medium-size datasets** ($500 \leq N < 5000$)**.** Points/stars denote median ranks across datasets, with horizontal bars showing 95% bootstrap confidence intervals for the median rank.

### G.4. RQ4: Compare with General Tabular Foundational Models

This section compares the SurvivalPFN, StaticSurvialTFM with other general TFMs. Specifically, we includes the most advanced TFMs including: TabPFN v2.5 (Grinsztajn et al., 2025), TabICL v2 (Qu et al., 2026), MITRA (Zhang et al., 2025), and TabDPT (Ma et al., 2024) regressors.

For the these TFM regression baselines, we train only on uncensored training examples because these models cannot natively deal with right-censoring datasets. TabPFN and TabICL are used as quantile regressors: they predict event-time quantiles, which are monotonized and converted into survival curves for distributional evaluation. MITRA and TabDPT are used as point regressors: they predict a single event time per test subject (just like cSVR). Since point predictions do not define a full survival distribution, distributional metrics such as IBS and D-calibration are not calculated and ranked as the worst among all the methods, while CI, MAE, and log-rank style comparisons are computed from the predicted event times.

For StaticSurvialTFM (Kim et al., 2026), it is a static fomula that can convert any classifier to survival predictor. We instantiate this static formulation with TabDPT and MITRA classifier backbones, predict failure probabilities over the cutoff grid, convert them to survival probabilities, and enforce monotone survival curves. We choose TabDPT to match with the model architecture of SurvivalPFN (for a fair comparison). We include MITRA as it is the best performing backbone described in Kim et al. (2026).

The results present here uses the same evaluation protocol as described in Appendix G.1. Figure 16 includes both general TFMs and the StaticSurvivalTFM wrapper instantiated with TabDPT and MITRA. Overall, SurvivalPFN remains the strongest and most consistent method: it achieves the best aggregate rank and ranks first or near-first across nearly all metrics. The largest gains appear for IBS and D-calibration, where SurvivalPFN clearly outperforms both direct TFM

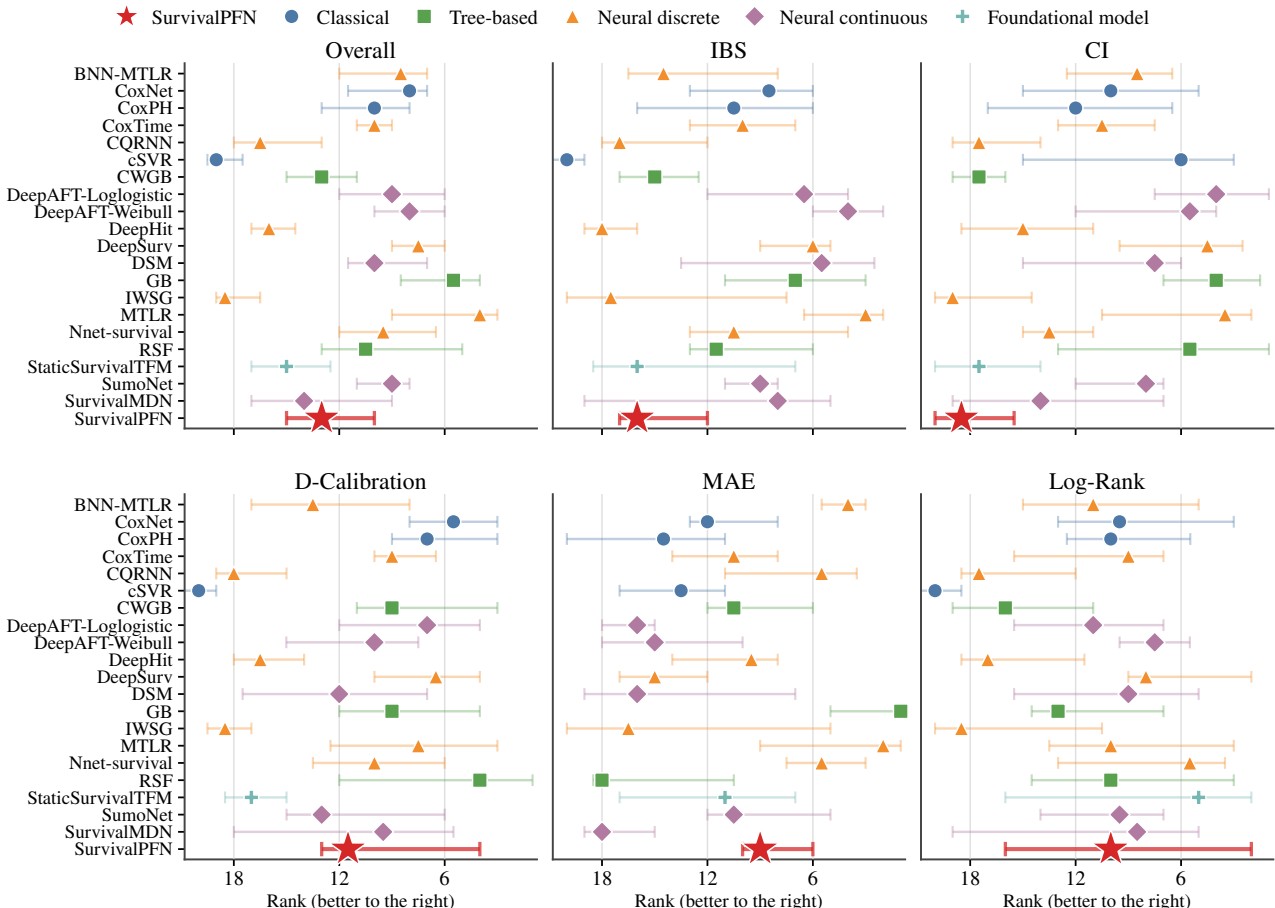

*Figure 11.* **Model ranks across 10 large-size datasets** ($N \geq 5000$). Points/stars denote median ranks across datasets, with horizontal bars showing 95% bootstrap confidence intervals for the median rank.

baselines and StaticSurvivalTFM variants, indicating better probabilistic survival estimation and calibration. SurvivalPFN also performs best on CI and log-rank, showing that its advantage extends beyond distributional accuracy to risk ranking and group separation.

The StaticSurvivalTFM performance is really sensitive to the backbone TFM model – which aligns the findings in (Kim et al., 2026). Using MITRA as the backbone improve over using TabDPT, especially for CI and log-rank, confirming that survival-specific label construction is helpful. However, their performance is less stable across metrics: StaticSurvivalTFM (MITRA) is competitive on MAE, CI, but does not match SurvivalPFN on IBS, D-calibration and Log-rank; StaticSurvivalTFM (TabDPT) performs well on log-rank but is weaker on other. In contrast, the direct regression baselines – TabPFN, TabICL, MITRA, and TabDPT – are consistently worse, despite being strong general tabular predictors.

These results support the main conclusion of **RQ4**: survival prediction benefits from a foundation model trained with survival-specific supervision and censoring-aware synthetic tasks, rather than relying only on generic tabular in-context learning.

### G.5. RQ5: Ablation Studies

**SurvivalPFN Ablation Configurations.** Table 5 summarizes the SurvivalPFN variants used in the ablation study. Each row corresponds to one pretrained checkpoint and is defined by six configuration choices.

**Predictive Pretraining** specifies whether the model is initialized from the predictive PFN-style pretraining phase before survival-specific training. ✓ means that the model first undergoes the general predictive pretraining stage described in Appendix D.2, and is then further trained with the survival phase. A value of ✗ means that survival-phase training starts

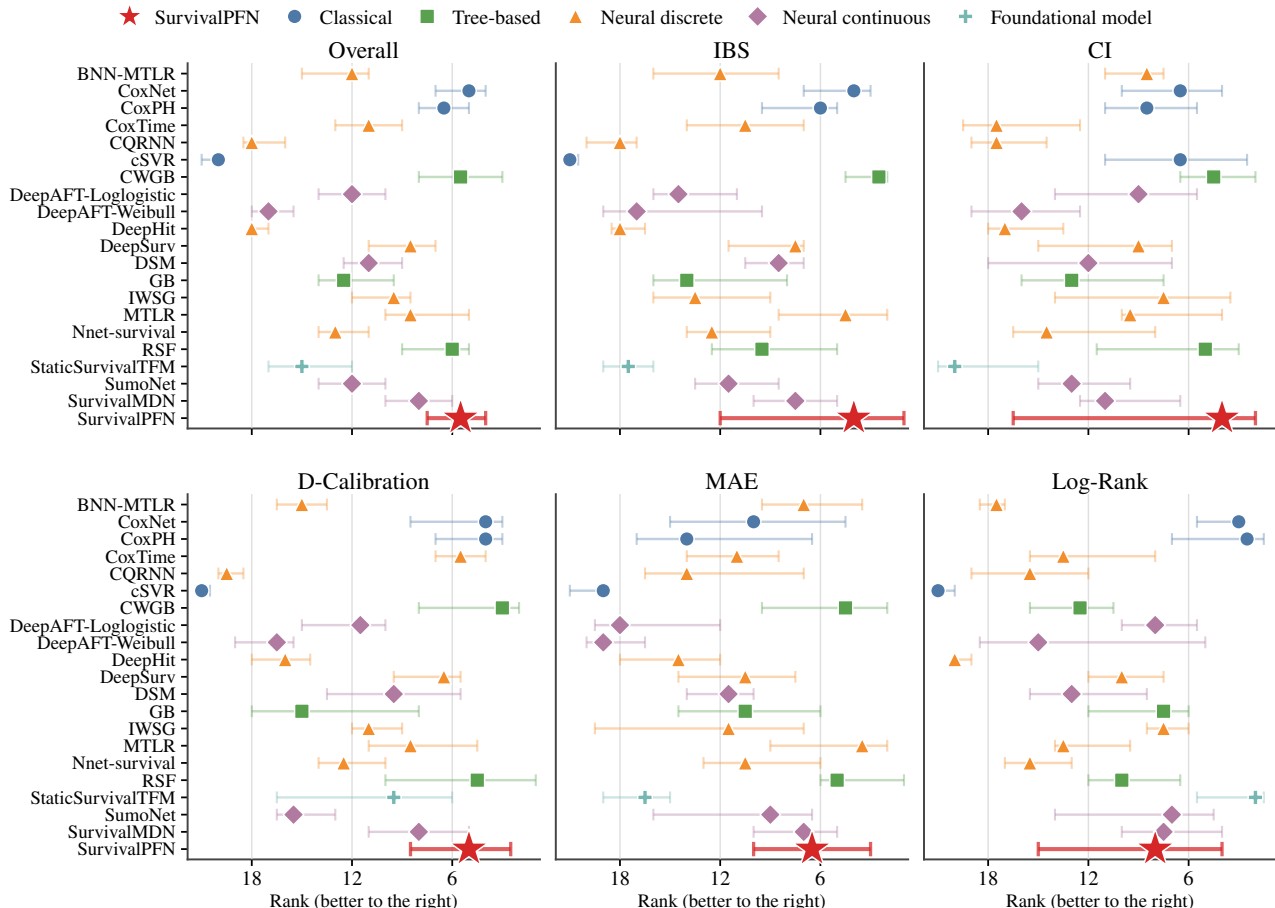

*Figure 12.* **Model ranks across 12 low-censoring-rate datasets (censoring rate $< 33\%$).** Points/stars denote median ranks across datasets, with horizontal bars showing 95% bootstrap confidence intervals for the median rank.

without this predictive initialization. This ablation tests whether generic PFN-style in-context predictive pretraining improves downstream survival prediction.

**Prior** specifies the synthetic survival prior used to generate the right-censored pretraining tasks, as described in Appendix D.1. We consider four possible prior families – the *naive prior*, the *survival-distribution prior*, the *mixture prior* and the *kitchen-sink prior*.

**Time transformation** specifies the monotone transformation applied to event and censoring times before discretization, as described in Appendix D.3. We try the `lognormal2normal` and the `time2quantile` transformations.

**Loss** specifies how the discretized predictive distribution is trained in transformed-time space, as described in Appendix D.4. The *NLL* setting uses a one-hot discrete negative log-likelihood, assigning all target mass to the bin containing the latent target time. The *CE* setting uses the smoothed histogram cross-entropy loss, where the latent target time is converted into a narrow Gaussian-smoothed histogram over bins.

**Query schedule** specifies how the query indicator $\widetilde{\delta}^*$ is selected during training, following Section 3. We try the *event-only*, *both*, and *random* strategies.

**Variable train ratio** specifies whether the ratio between context/training samples and query/inference samples is varied during synthetic pretraining. A value of ✓ means that this ratio is randomized across synthetic tasks, exposing the model to different amounts of context information and encouraging robustness to varying downstream train/test splits. A value of ✗ means that the ratio is fixed at 70%/30% during training.

Together, these choices define a full configuration space of $2 \times 4 \times 2 \times 2 \times 3 \times 2 = 192$ possible variants (corresponding

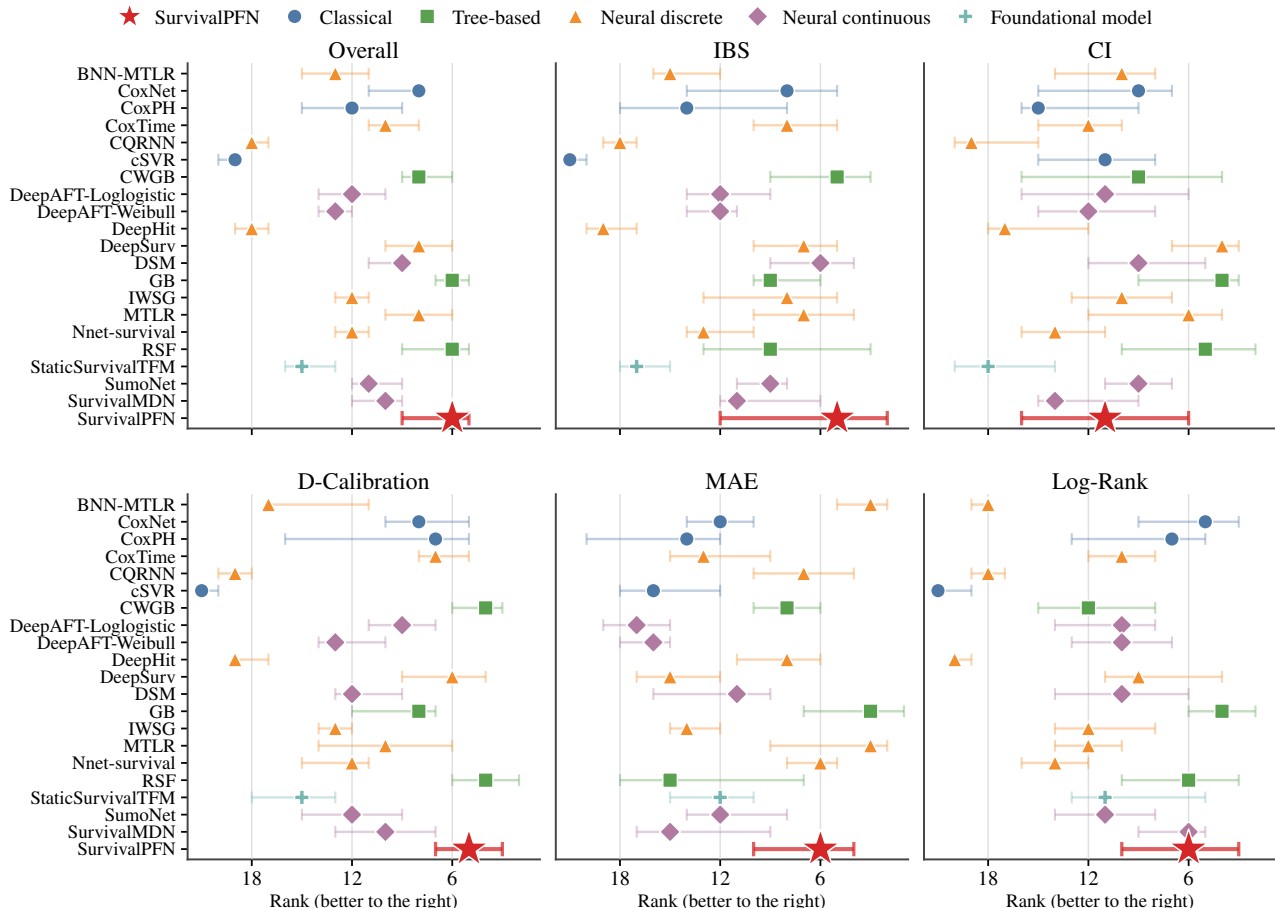

*Figure 13.* **Model ranks across 25 medium-censoring-rate datasets (censoring rate $\geq 33\%$ and $< 67\%$).** Points/stars denote median ranks across datasets, with horizontal bars showing 95% bootstrap confidence intervals for the median rank.

respectively to predictive pretraining, prior family, time transformation, loss, query schedule, and variable train ratio). Exhaustively training all variants would be computationally expensive, so we selectively evaluate the representative configurations in Table 5. The model marked with † is the best validation run and is used as the default SurvivalPFN checkpoint elsewhere in the paper.

**Ablation results.** Figure 17 summarizes the rank of each SurvivalPFN configuration. The selected checkpoint, v01, achieves the strongest overall behavior: it attains the best median rank across metrics among the evaluated configurations, and is particularly strong on distributional metrics, ranking best on IBS and D-calibration. This configuration uses predictive pretraining, the survival-distribution prior, the `lognormal2normal` transformation, the one-hot NLL objective, the Both query schedule, and a fixed train/query ratio. Its strong IBS and D-calibration performance suggests that this combination is especially effective for learning calibrated posterior predictive survival distributions, which is the primary target of SurvivalPFN.

Several trends emerge from the ablation. First, the survival-distribution prior is consistently stronger than the naive prior under otherwise similar settings. This suggests that directly modeling flexible positive-time distributions provides a more useful synthetic pretraining signal than treating generic tabular outputs as raw survival times. The kitchen-sink prior performs competitively but does not clearly dominate the survival-distribution prior, this might indicating that simply increasing prior diversity is not sufficient; the match between the prior family and the survival-prediction target also matters.

Second, the `lognormal2normal` transformation is preferred in this set of experiments. The clearest comparison is between v01 and v02, replacing `lognormal2normal` with `time2quantile` substantially worsens the overall rank and degrades all five metric-specific ranks. This pattern suggests that the smooth positive-time coordinate and tail extrapolation

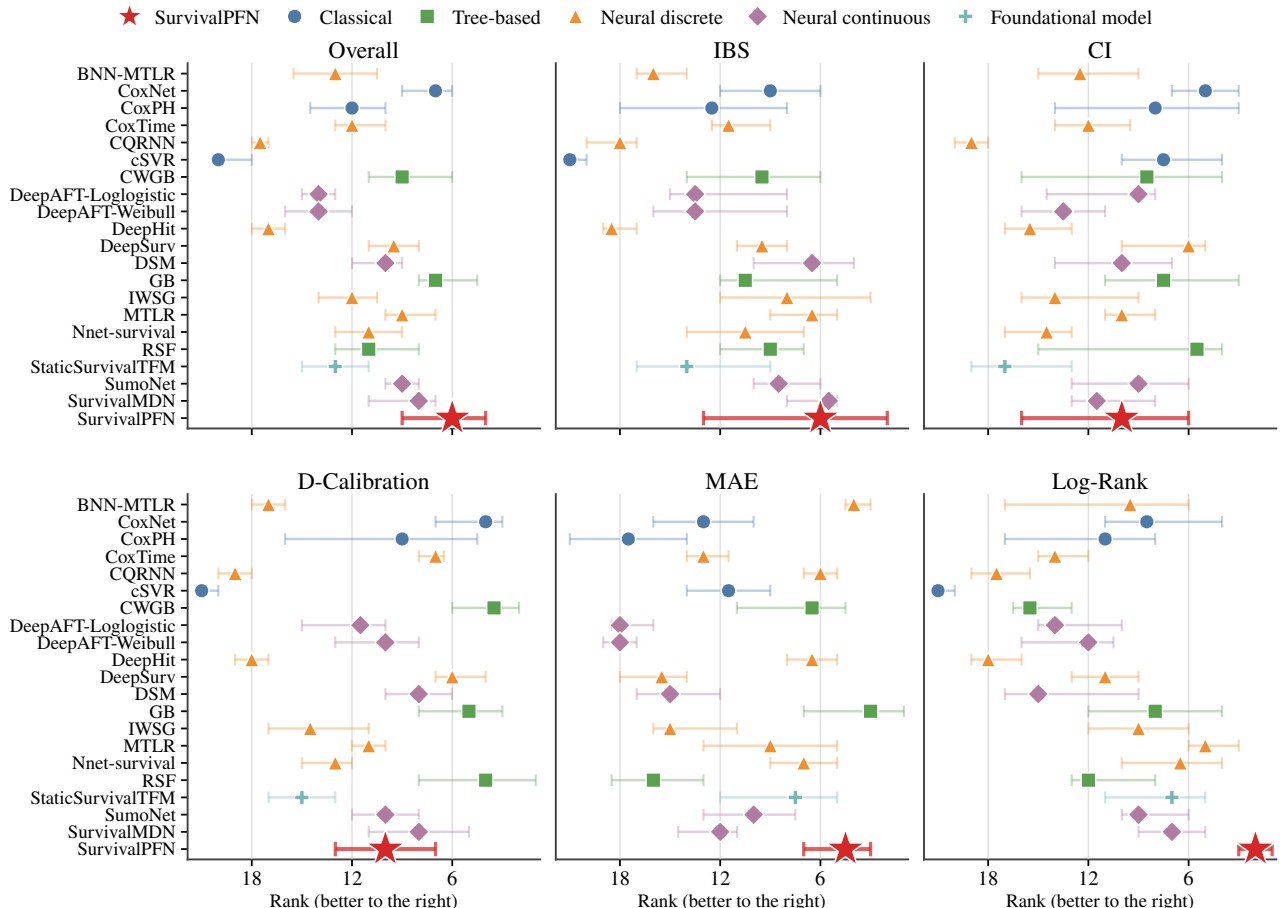

*Figure 14.* **Model ranks across 24 high-censoring-rate datasets (censoring rate $\geq 67\%$).** Points/stars denote median ranks across datasets, with horizontal bars showing 95% bootstrap confidence intervals for the median rank.

provided by `lognormal2normal` are useful for transferring across heterogeneous real-world time scales, whereas the empirical quantile transform may lose information about tail behavior.

Finally, predictive pretraining is generally helpful but not uniformly decisive in this limited ablation set. Similarly, varying the train/query ratio does not show a clear monotonic benefit in the evaluated subset.

## H. Related Work

**Classical and Deep Survival Analysis.** A broad range of estimators has been developed for right-censored data. CoxPH (Cox, 1972) relies on proportional hazards and linear covariate effects, while fully parametric models such as exponential (Mendenhall & Hader, 1958) and Weibull (Peto & Lee, 1973) models impose explicit distributional forms. Modern machine-learning methods improve flexibility, but introduce other assumptions. Neural Cox-based models, such as DeepSurv (Katzman et al., 2018) and CoxTime (Kvamme et al., 2019), relax linear covariate effects but retain Cox-style hazard modeling. Discrete-time models, including MTLR (Yu et al., 2011; Fotso, 2018) and DeepHit (Lee et al., 2018; 2019), depend on a chosen time grid and may become overparameterized with many bins. Continuous-time neural models are more flexible, but still impose structure through parametric mixtures (Nagpal et al., 2021), latent-variable models (Ranganath et al., 2016; Miscouridou et al., 2018), monotonic density estimators (Rindt et al., 2022; Chilinski & Silva, 2020), or neural ODE hazards (Groha et al., 2020; Tang et al., 2022).

**Bayesian Survival Analysis.** Bayesian survival models quantify uncertainty by placing priors over model parameters and integrating over posterior uncertainty. Recent neural variants include BNN-ISD, which uses Bayesian neural networks

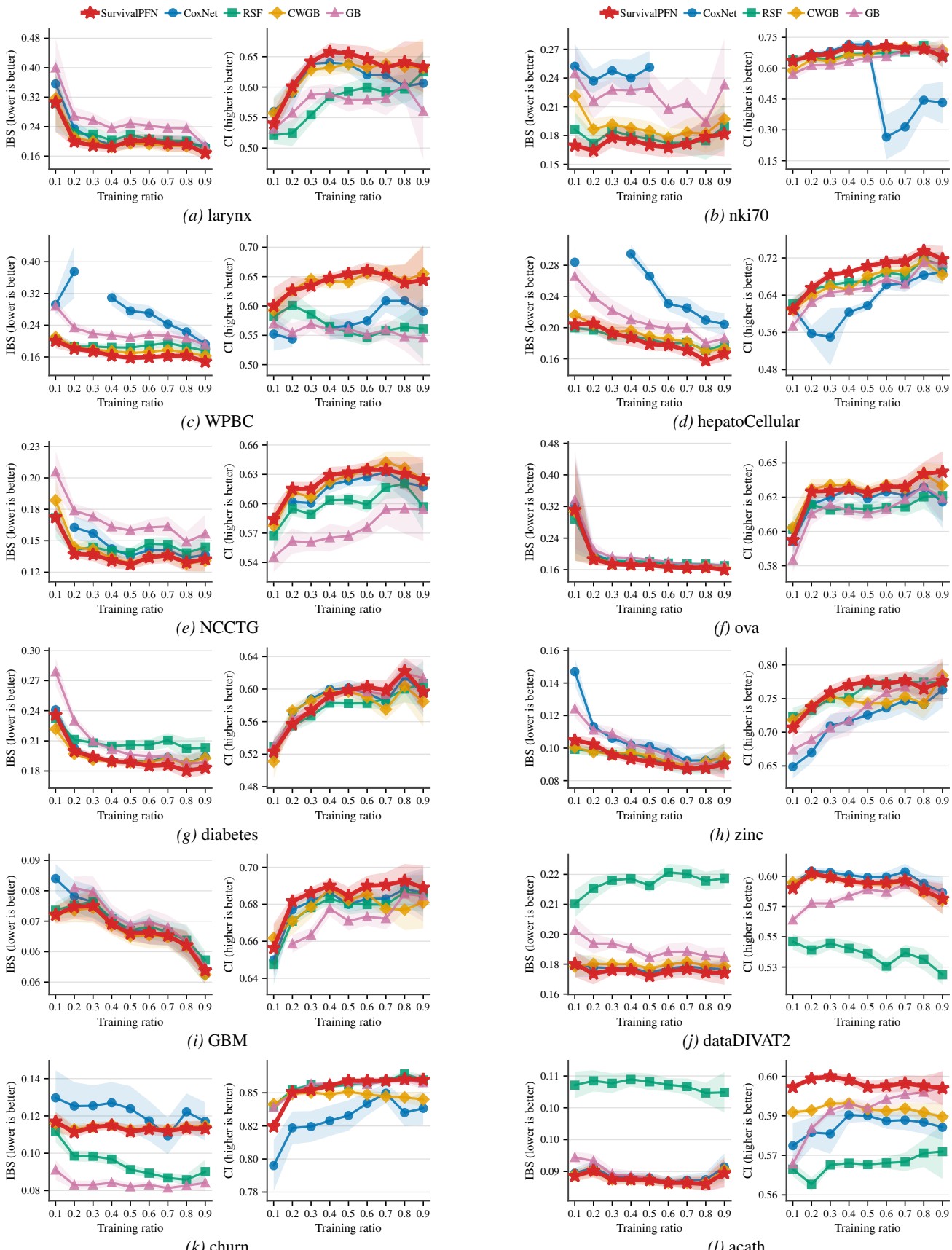

*Figure 15.* Sensitivity to the training/context ratio across selected 16 datasets.

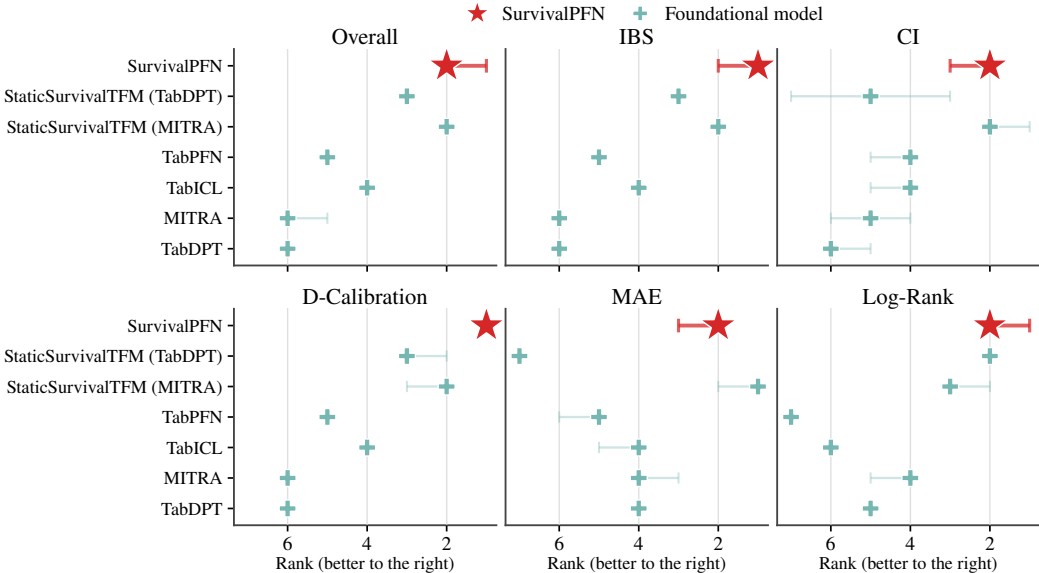

*Figure 16.* **Compare SurvivalPFN with general TFMs across 61 benchmark datasets.** Plotting conventions follow Figure 5.

to obtain credible intervals and supports feature selection (Qi et al., 2023b); Bayesian LSTM-SURV, which combines the survival likelihood with Bayesian mixed-effects updating under a Weibull parametric form (Gao et al., 2025); and NeuralSurv, which uses variational inference for Bayesian deep survival prediction (Monod et al., 2025). While these methods demonstrate the value of Bayesian uncertainty quantification, they remain tied to method-specific assumptions and per-dataset inference or updating. SurvivalPFN instead amortizes posterior-predictive survival inference during prior-data pretraining, producing survival distributions for a new right-censored dataset with a single forward pass and without dataset-specific posterior sampling, variational optimization, or hyperparameter tuning.

**Tabular Foundation Models for Survival Analysis.** Concurrent work has begun to adapt tabular foundation models to survival prediction. Kim et al. (2026) convert survival prediction to a sequence of binary classification tasks over discretized time points, enabling off-the-shelf TFMs without survival-specific pretraining. Seletkov et al. (2026) instead pretrain a survival-specific in-context model on synthetic right-censored datasets from parametric extended-hazard mechanisms. SurvivalPFN differs by using a broader family of identifiable right-censored DGPs, supporting covariate-dependent censoring under conditional independence, avoiding explicit parametric survival families, and providing a posterior-predictive consistency argument. Appendix I gives a more detailed comparison with these concurrent approaches.

## I. Concurrent Works on Tabular Foundation Models for Survival Analysis

We discuss two concurrent approaches that adapt TFMs to right-censored survival prediction.

**Classification-Based Framework with Off-the-Shelf TFMs.** Kim et al. (2026) propose a conversion from survival analysis to binary classification, allowing existing TFMs to be used without survival-specific pretraining. Given predefined discretization points

$$0 = t_0 < t_1 < \cdots < t_{K-1},$$

they define time-indexed binary labels

$$Y_{i,k} = \mathbb{1}(T_i \leq t_k),$$

so that each original tuple $(x_i, t_i, \delta_i)$ is expanded into multiple classification examples indexed by $k$. Under right censoring, labels after the censoring time are treated as missing; equivalently, their binary cross-entropy objective is evaluated only when $t_k < C_i$. Under conditional independent censoring and positivity, they show that minimizing the population binary cross-entropy loss recovers the true failure probabilities,

$$p(x, t_k) = \Pr(T \leq t_k \mid X = x),$$

*Table 5.* **SurvivalPFN ablation configurations.** Each row corresponds to one pretrained SurvivalPFN variant. The internal checkpoint-path column from the experiment log is omitted. "Surv.-dist." denotes the survival-distribution prior. NLL denotes the one-hot discrete negative log-likelihood over transformed-time bins; CE denotes the smoothed histogram cross-entropy loss.

| Model | Predictive Pretrain | Prior | Time Transformation | Loss | Query Schedule | Variable Train Ratio |
|---|---|---|---|---|---|---|
| v01[†] | ✓ | Surv.-dist. | `lognormal2normal` | NLL | Both | ✗ |
| v02 | ✓ | Surv.-dist. | `time2quantile` | NLL | Both | ✗ |
| v03 | ✗ | Surv.-dist. | `time2quantile` | CE | Random | ✓ |
| v04 | ✓ | Surv.-dist. | `time2quantile` | CE | Random | ✓ |
| v05 | ✓ | Surv.-dist. | `time2quantile` | NLL | Random | ✓ |
| v06 | ✓ | Surv.-dist. | `lognormal2normal` | NLL | Random | ✓ |
| v07 | ✓ | Kitchen-sink | `lognormal2normal` | CE | Random | ✓ |
| v08 | ✓ | Kitchen-sink | `lognormal2normal` | NLL | Event-only | ✓ |
| v09 | ✓ | Surv.-dist. | `lognormal2normal` | CE | Random | ✓ |
| v10 | ✓ | Surv.-dist. | `lognormal2normal` | NLL | Event-only | ✓ |
| v11 | ✓ | Naive | `lognormal2normal` | CE | Random | ✓ |
| v12 | ✓ | Naive | `lognormal2normal` | NLL | Event-only | ✓ |
| v13 | ✗ | Naive | `lognormal2normal` | CE | Random | ✓ |
| v14 | ✗ | Naive | `lognormal2normal` | NLL | Event-only | ✓ |

[†]Best validation run; this checkpoint is used as the default SurvivalPFN model elsewhere in the paper.

and hence the survival probabilities $S(t_k \mid x) = 1 - p(x, t_k)$. This formulation is attractive because it can immediately use strong off-the-shelf TFMs such as MITRA, TabPFN v2.5, and TabICL v2 (Zhang et al., 2025; Grinsztajn et al., 2025; Qu et al., 2026), without retraining a survival-specific model.

The main limitation is that this reduction increases the effective context size. Each subject produces up to $K - 1$ time-indexed classification examples, so a dataset with $N$ subjects becomes an expanded context of order $N(K - 1)$. This is manageable for small datasets, but can exceed the input limits of current TFMs on medium or large survival datasets, requiring subsampling and potentially discarding observed survival information. The method also inherits limitations of discrete-time classification, including dependence on the time grid and the need for post-hoc monotonicity correction of predicted survival curves. In our experiments, we include the static version of this approach as StaticSurvivalTFM.

**Survival-Specific Prior-Fitted In-Context Learning.** Seletkov et al. (2026) propose Survival In-Context (SIC), a survival-specific prior-fitted model trained on synthetic right-censored datasets. Their data generator first samples covariates and latent risk variables $(\eta_1, \eta_2)$ from structural causal models (SCMs). Event times are then generated using the extended-hazard model

$$h(t \mid x) = h_0(te^{\eta_1})e^{\eta_2},$$

which yields

$$T = e^{-\eta_1} H_0^{-1}\big(e^{\eta_1 - \eta_2}(-\log U)\big), \qquad U \sim \mathrm{Unif}(0, 1),$$

where $H_0^{-1}$ is chosen from a set of parametric baseline families such as Weibull, lognormal, log-logistic, Gompertz, and Birnbaum-Saunders distributions. Censoring is generated by random censoring assumption – not dependent on covariates and event times.

Architecturally, SIC builds on TabICL, adds a time-event embedding, and uses a DeepHit-style discrete-time survival head Lee et al. (2018) trained with a likelihood-plus-ranking loss.

SIC is closely related to SurvivalPFN in that both methods pretrain an in-context model specifically for survival prediction. However, the two approaches differ substantially in prior design, which is central to the PFN paradigm. SIC's prior is based on a parametric extended-hazard construction and only random censoring, whereas SurvivalPFN uses a broader family of identifiable right-censored DGPs, including random censoring and covariate-dependent censoring mechanisms satisfying conditional independence. SurvivalPFN also avoids committing to explicit parametric hazard or survival families, allowing the event and censoring distributions to be generated by more flexible stochastic neural mechanisms. Finally, SurvivalPFN

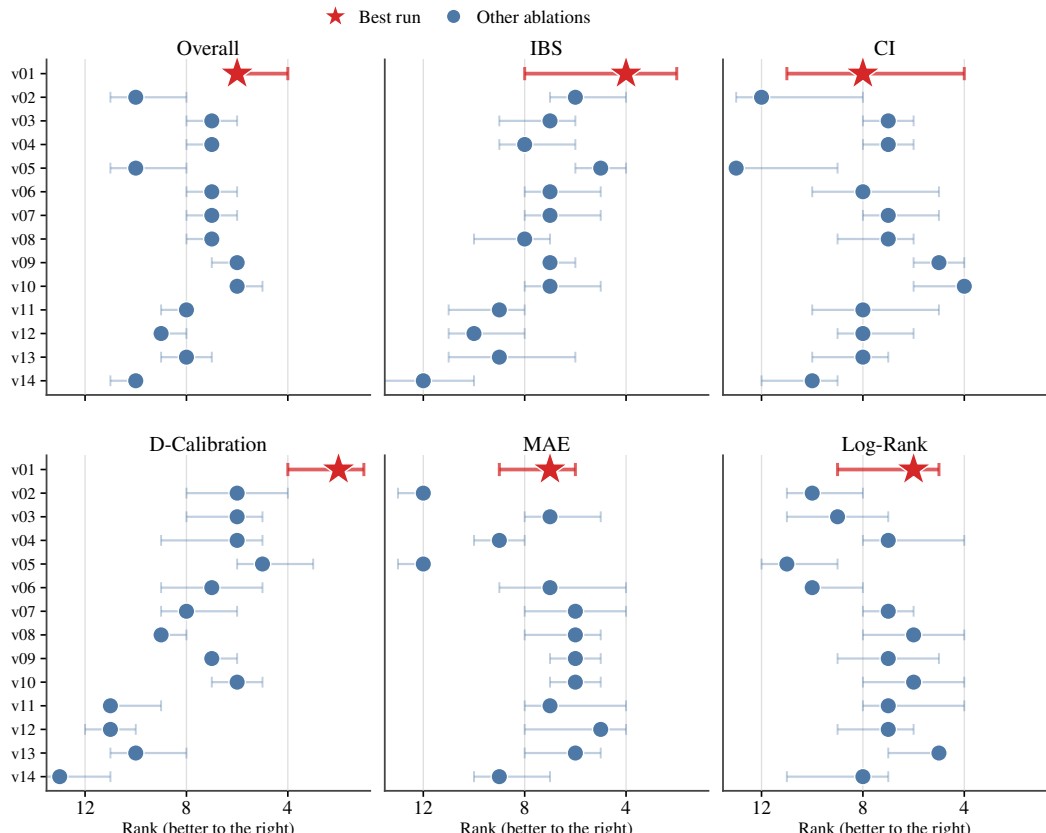

*Figure 17.* Ablation study over SurvivalPFN training configurations. Each row corresponds to one pretrained SurvivalPFN variant, with v01 marked as the selected best validation run and used as the default checkpoint elsewhere in the paper. Plotting conventions follow Figure 5.

is accompanied by a posterior-predictive consistency guarantee for identifiable survival priors, whereas SIC only provides empirical evidence.

The empirical scope also differs: SIC evaluates on a smaller benchmark with one main metric and a limited set of baselines, while our study evaluates on 61 held-out datasets, five metrics, and 21 baselines. Since SIC has not released public model weights or code, we cannot include it in our direct empirical comparison.

