# OpenReview forum: "SurvivalPFN: Amortizing Survival Prediction via In-Context Bayesian Inference"
_ICML.cc/2026/Workshop/FMSD — FMSD @ ICML 2026 SpotlightOral_

### Official Review · Reviewer_L7Bm · 2026-05-13

**Rating:** 8
**Confidence:** 4

**Review:**

**Summary**: The work introduces a prior-data fitted network that is pretrained on millions of synthetic, identifiable, right-censored data-generating processes. The method amortizes Bayesian inference through in-context learning, shifting the computational cost from per-dataset fitting to a single pretraining stage.


**Strengths**:

- The proposed model obtains the best median performance and second-best runtime compared to all baselines in Figure 1.
- The proposed model is pretrained on millions of synthetic, identifiable, right-censored data-generating processes.
- The model returns a posterior predictive survival distribution rather than just regressed survival times.
- The contributions are outlined clearly in the introduction.
- The evaluation is comprehensive, consisting of 61 held-out survival datasets and 21 baselines.
- The discussion of how survival times and censoring are incorporated into the model is clear.
- The work reports five evaluation metrics.
- Figure 4 is an informative summary of the datasets.
- Evaluation includes error bounds across 10 repeated runs.


**Areas for Improvement**:

- Figure 1: suggest reversing the rank axis (lower is better) to align with the convention that lower prediction error is better. Figure 5: the rank axis should similarly be reversed to count upward, as the current decreasing direction is counterintuitive.

- A spider diagram summarizing model performance across dataset characteristics, such as dataset size, censoring rate, and survival tail, as outlined in Figure 4, may provide a useful visual summary of where each model performs well or poorly. This would provide a visual to align with conclusions drawn: “SurvivalPFN’s strongest relative gains occur on smaller datasets” (lines 150-151).

- Figure 2 would benefit from a more formal presentation.

**Justification for Score**: This is one of the most comprehensive studies I'm aware of for PFN-based survival regression and the results are very promising for the proposed approach. I think researchers working with Tabular Foundation Models and Survival Regression would be very interested in this paper.

---

### Official Review · Reviewer_G9Co · 2026-05-22
**Review of SurvivalPFN: Amortizing Survival Prediction via In-Context Bayesian Inference**

**Rating:** 8
**Confidence:** 3

**Review:**

## Summary
Authors introduce SurvivalPFN, a model pre-trained on synthetic data in a similar spirit to TabPFN, but focusing on survival analysis, where the data is often right-censored, i.e. the event time is oftentimes not recorded. In the previously existing approaches, censoring poses a significant modelling task, requiring an expert to tune the models to each dataset separately. SurvivalPFN tries to address this by exposing the transformer model to multiple synthetic right-censored data generating processes during pre-training, trying to achieve generalizability by the model. The processes are identifiable by assuming (for artificial data: enforcing) conditional independence of the event time and censoring time. Under this assumption, the authors show that the model is both theoretically sound and empirically very strong -- the comparison is done on multiple (61) datasets and against multiple baselines.

## Strengths
* Extending the PFN idea to a new domain, which can't be easily treated with off-the-shelf solutions. The approach promises a new status quo for the domain, with significantly reduced expert manual labor.
* I feel that the authors make a good job with introducing the overall concept and main modelling assumptions of SurvivalPFN with text and figures, especially in the context of the Bayesian formulation which is sometimes not too "reader friendly".
* Empirical results are over multiple datasets, multiple metrics, and compared against multiple baselines.
* Strong empirical performance across the board (leading in some metrics, while being competitive in the others, also the inference speed is very good.

## Areas for Improvement and Comments
* Conditional independence assumption is obviously somewhat limiting the applicability in some domains, as noted in the last section.
* The authors mention that long context is an issue, as typical for PFN-type models, but don't elaborate more. E.g., were there datasets which couldn't be used, or where performance significantly degraded?
* It's more of a question to the session chairs, but I feel like the manuscript pushes the idea of 4-page main part + additional material in the appendices quite to the limit. One example: "Additional results are deferred to the appendix: stratified performance by sample size and censoring rate appears in Appendix G.1; training-set-size sensitivity appears in Appendix G.3; comparison with general-purpose tabular foundation models appears in Appendix G.4; and ablations in Appendix G.5." At some point one might get the whole 4-page text being references to the appendices...
* Figure 15 is cut off in my version, and overall I'd argue it would be advisable to have some table with numerical values for the metrics, at least as a supplement or a link, to allow easier verification of the numbers by independent researchers, as the median rank is a rather indirect way of showing performance.